# PEVLM: Parallel Encoding for Vision-Language Models

## Abstract

Vision-Language Models (VLMs) have demonstrated strong capabilities in multimodal understanding and generation tasks. However, their application to long video understanding remains hindered by the quadratic complexity of standard attention mechanisms. In this work, we introduce **PEVLM**, a fine-tuning-free parallel encoding method designed to enhance the prefilling efficiency of VLMs in long video scenarios. To the best of our knowledge, this is the first work to adapt parallel encoding to VLMs. PEVLM partitions the input video into context blocks with a shared sink block, while preserving sequential position embeddings to align the attention score distribution with that of Full-Attention. This design reduces the complexity of attention from $O((T \times N)^2)$ to $O(T \times N)$ where $T$ is the number of frames and $N$ the number of tokens per frame, with minimal loss in accuracy. Extensive experiments across multiple state-of-the-art models and benchmarks demonstrate that PEVLM consistently outperforms existing parallel encoding approaches, achieving up to **7.47x** speedup in attention computation and reducing end-to-end latency by **44%** to **50%**. Remarkably, PEVLM not only maintains high accuracy, but in some settings even surpasses Full-Attention performance. Under strict latency constraints, it achieves substantial gains, improving accuracy from **23.26%** to **61.03%**. These results underscore the effectiveness of PEVLM for low-latency, long-context video understanding, making it a promising solution for real-world applications.

## 1 Introduction

In recent years, Vision-Language Models (VLMs) have become a central research focus at the intersection of computer vision and natural language processing. These models have demonstrated impressive performance in a wide range of multimodal understanding and generation tasks (Alayrac et al., 2022; Li et al., 2020; Zhang et al., 2024c; Chen et al., 2024; Bai et al., 2025). As their capabilities continue to improve, VLMs are being applied in increasingly complex domains, including robotics (Black et al., 2024; Luo et al., 2025; Team et al., 2025), autonomous driving (Gao et al., 2024; Hu et al., 2023; Wang et al., 2024), and healthcare (Liu et al., 2024). These application scenarios often demand processing longer video inputs.

However, a key obstacle to applying VLMs to long-video inputs is the quadratic complexity of transformer attention during the prefilling stage (Vaswani et al., 2017; Beltagy et al., 2020). To address this issue, a widely adopted technique in large language models (LLMs) is the parallel encoding mechanism (Li et al., 2024b; Acharya et al., 2025; Ma et al., 2025; Ratner et al., 2023; Yang et al., 2025; Yen et al., 2024; Lu et al., 2024). In this approach, the input context is divided into multiple blocks, each block independently encoded into Key-Value (KV) states, thus reducing the computational complexity from $O(L^2)$ to $O(L)$. Moreover, parallel encoding alleviates the issue of "lost in the middle phenomenon" by reducing the number of tokens participating in the softmax operation and can even achieve accuracy surpassing that of Full-Attention in some scenarios (Liu et al., 2023; Yang et al., 2025; Veličković et al., 2025).

Given this context, a natural question arises: Can existing parallel encoding methods for LLMs be directly applied to VLMs for faster inference and deployment? In Table 1, we evaluate several state-of-the-art models on widely used video benchmarks. Although we observe significant accuracy drops with these methods, especially for models in the Qwen-VL family and the InternVL3_5 family.

In extreme cases, the output becomes empty or garbled, with accuracy falling to 0%. To investigate this, we analyze attention score distributions and find a misalignment between Full-Attention and parallel encoding. Based on this insight, we propose PEVLM (Parallel Encoding for Vision-Language Models—a new attention mechanism tailored for VLMs). PEVLM requires no fine-tuning, introduces no additional parameters, and involves minimal code changes, making it a lightweight solution to accelerate long-video processing in both cloud and edge settings. Our contributions are as follows.

- We systematically analyze the distribution characteristics of attention scores under parallel encoding in VLMs, identifying a key reason for the misalignment between the attention score distributions of parallel encoding and Full-Attention: reusing position embeddings across blocks leads to the loss of critical video information.
- We propose PEVLM to recover the accuracy of parallel encoding by applying three alignment steps: (i) segmenting contexts into blocks by video frames rather than tokens; (ii) using the system prompts and the initial frames of the video as Sink Block for all blocks to avoid the duplication of abnormal distribution of initial tokens; (iii) preserving the sequential position embeddings instead of reusing position embeddings across blocks. With these alignment strategies, PEVLM reaches higher accuracy than existing parallel encoding methods, and reaches 98.24% to 104.80% of the accuracy of Full-Attention on different models.
- PEVLM reduces the computational complexity of the attention mechanism from $O((T \times N)^2)$ to $O(T \times N)$, where $T$ is the number of frames and $N$ the number of tokens per frame. (i) For 100k (text&video) token contexts prefilling, the attention layer achieves a 7.47× speedup, while the end-to-end 2.58x speedup without compromising generation quality. This highlights the practical viability and superiority of PEVLM in cloud deployment scenarios. (ii) Under fixed latency constraints, PEVLM increases accuracy from 23.26% to 61.03%, showcasing its critical value for latency sensitive applications.

## 2 OBSERVATIONS

To appropriately adapt the parallel encoding method for VLMs, in this section, we first review the foundational components of the standard attention mechanism, and then highlight the key differences between its application in LLMs and VLMs. These differences explain why off-the-shelf LLM parallel encoding methods underperform in long-video VLMs, motivating the design of PEVLM.

### 2.1 STANDARD ATTENTION MECHANISM

The softmax attention mechanism (Vaswani et al., 2017) serves as the core of transformer-based models. For a given query $Q \in \mathbb{R}^{n \times d}$, keys $K \in \mathbb{R}^{m \times d}$, and values $V \in \mathbb{R}^{m \times d}$, the output is computed as:

$$O = \text{Softmax} \left( \frac{QK^T}{\sqrt{d}} \right) V, \tag{1}$$

where $d$ is the model dimension.

Two key phenomena influence attention behavior in long sequences. First, the attention sink effect (Lab & AI, 2023; Sun et al., 2024) causes tokens at the input's beginning, often the BOS token or system prompt, to consistently receive disproportionately high attention scores, anchoring the model's focus. Second, position embeddings encode token order, enabling the model to discern sequential relationships. During prefilling, the computational cost is $\text{OP}_{\text{Attn}} = 2HL^2$, yielding $O(L^2)$ complexity, where $L$ is the context length and $H$ the hidden size. This quadratic scaling severely bottlenecks inference for long inputs like extended videos.

### 2.2 KEY DIFFERENCES: LLMS VS. VLMS

**Attention Sink** APE (Yang et al., 2025) and Star-Attention (Acharya et al., 2025) have shown that sharing a common attention sink across parallel encoding blocks significantly enhances the accuracy of LLMs. **However, it has been observed that attention sinks are present not only in the initial system prompt, but also in the video frames in VLMs** (Huang et al., 2024; Zhang et al., 2024b; Kang et al., 2025). Drawing on the experience of applying attention sinks in LLMs, achieving optimal performance in VLMs may require incorporating a sufficient number of frames into the sink.

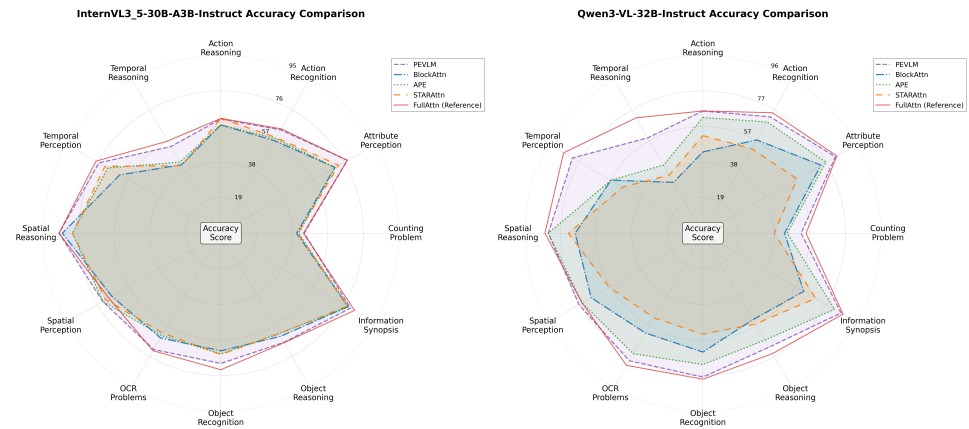

Figure 1: Radar-chart comparison of attention mechanisms (Full Attention, PEVLM, APE, BlockAttn, and StarAttn) on two representative large-scale models: InternVL3.5-30B-A3B (left) and Qwen3-VL-32B (right). PEVLM closely matches Full Attention across all Video-MME categories, while other methods (APE, BlockAttn, StarAttn) show noticeable degradation. More detailed results for all model scales are provided in Appendix G.

**Position Embedding** Recent efficient attention mechanisms, such as APE (Yang et al., 2025), Block Attention (Ma et al., 2025), and Star Attention (Acharya et al., 2025), often rely on reusing or sharing position embeddings across blocks to support parallel prefilling. However, these designs introduce significant limitations in video-centric tasks. In temporally sensitive categories and OCR-related tasks that require fine-grained spatial alignment across frames, these methods exhibit clear accuracy degradation compared to Full Attention (as shown in Figure 1). We find that the major performance drop arises from the reuse of position embeddings: compressing heterogeneous frame sequences into repeated positional templates disrupts the temporal identity of each frame, weakens cross-frame alignment, and distorts long-range dependency modeling.

**So, preserving the correct temporal order of video frames is critical for video understanding tasks. Blindly reusing position embeddings across blocks can disrupt the model's ability to capture temporal dependencies.**

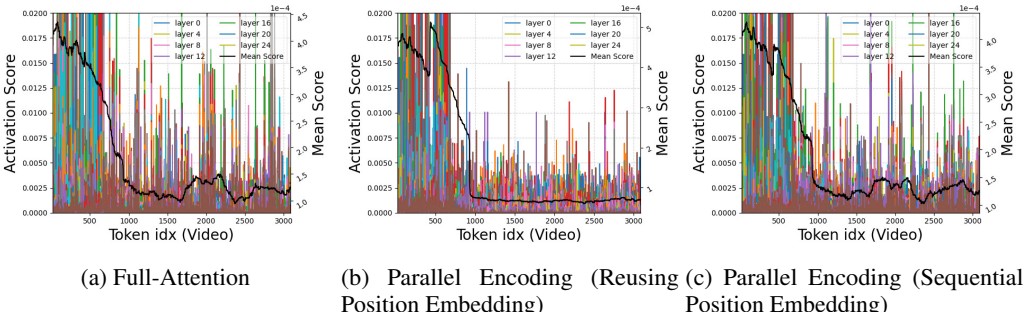

(a) Full-Attention     (b) Parallel Encoding (Reusing Position Embedding)     (c) Parallel Encoding (Sequential Position Embedding)

Figure 2: We examine the distributions of attention scores under Full Attention and Parallel Encoding using different positional embedding strategies, and further compute their moving averages (referred to as the mean score) to enable more systematic comparison. Due to space constraints, we present only the results on Qwen2.5-VL, more results are provided in Appendix H.

To support the above observations, we further collect the distribution of attention scores under Full-Attention and Parallel Encoding with different position embedding strategies, as shown in Figure 2, and summarize two key findings: (i) As shown in Figure 2(a), attention sinks are present not only in the initial system prompt, but also in the early video frames in VLMs. This suggests that it should be necessary to include early video frames when selecting sink tokens. (ii) As shown in Figure 2(b), compared to Full-Attention, the initial video frames receive even higher attention scores, while the remaining frames exhibit more uniform and lower attention scores. In contrast, when sequential

position embeddings are preserved, as illustrated in Figure 2(c), parallel encoding yields an attention score distribution that more closely resembles that of Full-Attention. These findings underscore the need for a VLM-specific parallel encoding strategy, which we introduce in the next section.

# 3 PEVLM

With the observations in the last section, we design PEVLM, which adaptively aligns the distribution of attention scores between Full-Attention and parallel encoding in VLMs, thereby boosting efficiency and performance.

## 3.1 PARTITIONING STRATEGY

As illustrated in Figure 3a, PEVLM partitioning strategy consists of three core steps:

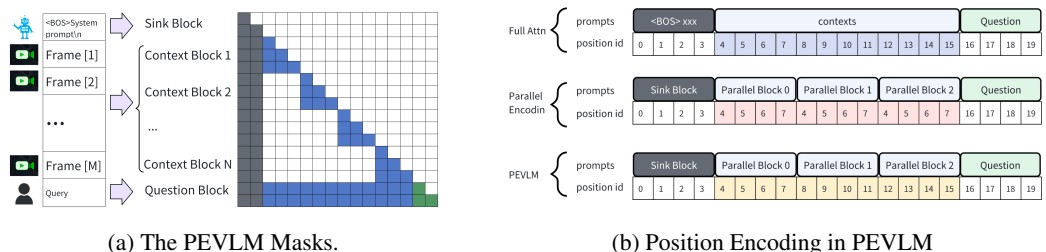

(a) The PEVLM Masks.
(b) Position Encoding in PEVLM

Figure 3: Illustration of Position Encoding and Masks in PEVLM

**Sink Block**: The initial text tokens (e.g., BOS token and system prompts) and the first several video frames are grouped into a dedicated Sink Block.

**Context Blocks**: The remaining video content is uniformly partitioned into Context Blocks by frames to reduce computational overhead and enable parallelization.

**Question Block**: The text tokens that follow the video input are left unsegmented as Question Block.

As described in the *Observations* section, similar to the case in LLMs, applying parallel encoding methods in VLMs suffers from the misalignment of attention score distributions when compared to Full-Attention. While prior work has attempted to address this issue by introducing additional hyperparameters to realign attention scores (Yang et al., 2025), such approaches are not ideal for efficient deployment. To address this, PEVLM draws inspiration from Block Attention (Ma et al., 2025) and TurboRAG (Lu et al., 2024), and retains sequential position embeddings, as shown in Figure 3b. In contrast to these methods, which dynamically adjust positional embeddings, often to accommodate context reuse through sequence reordering, PEVLM forgoes such updates. Since video frames are processed in their original temporal order, PEVLM directly applies the sequential position embedding to each visual token within context blocks, without modifying their positions after query concatenation.

By preserving sequential position embeddings, PEVLM maintains attention score distributions that closely resemble those of Full-Attention. It is worth noting that, unlike some parallel encoding methods in LLMs that can enhance a model's ability to process longer contexts (Ratner et al., 2023; Yang et al., 2025), PEVLM does not provide such benefits. Because PEVLM preserves the original positional embedding of every input token, unlike in LLMs, it cannot effectively extend a model's usable context length. For example, in the case of LongVILA-7B-256f, the optimal number of sampled frames is 256. Using a higher frame sampling rate does not improve the model's accuracy. For PEVLM, higher sampling rates (which correspond to longer sequence lengths) can yield better acceleration ratios; however, PEVLM does not improve model accuracy once the sampling rate exceeds the optimal 256-frame setting. Therefore, all subsequent accuracy and performance evaluations in following sections are conducted within each model's optimal frame sampling range. Although increasing the sampling frequency may superficially provide larger speedup, it brings no accuracy benefits and is thus unnecessary.

## 3.2 FORMULATIONS

The computational formulation of PEVLM is defined as:

$$\text{Attn}s = f(Q_s, K_s, V_s), \tag{2}$$

$$\text{Attn}_{c_i} = f(Q_{c_i}, K_{s+c_0+\cdots+c_{i-1}}, V_{s+c_0+\cdots+c_{i-1}}), \tag{3}$$

$$\text{Attn}_q = f(Q_q, K_{s+c_{\text{all}}+q}, V_{s+c_{\text{all}}+q}), \tag{4}$$

where $s$ denotes the Sink Block, $q$ denotes the Question Block, and $c_i$ denotes the $i$-th Context Block. Symbols $Attn_s$, $Attn_{c_i}$, and $Attn_q$ represent the attention outputs for the corresponding blocks. The function $f(\cdot)$ corresponds to the standard softmax attention defined in Equation (1), with $Q$, $K$ and $V$ denoting the query, key, and value matrices respectively.

The total computational operation (OP) count of PEVLM is:

$$\text{OP}_{\text{PEVLM}} = \text{OP}_{\text{Sink}} + N \times \text{OP}_{\text{ContextBlock}} + \text{OP}_{\text{Quest}}, \tag{5}$$

$$\text{OP}_{\text{Sink}} = 2HS^2, \tag{6}$$

$$\text{OP}_{\text{ContextBlock}} = 2HB(S + B), \tag{7}$$

$$\text{OP}_{\text{Quest}} = 2HQL, \tag{8}$$

where $S$ denotes the sink block size, $N$ denotes the context block number, $B$ denotes the context block size, $Q$ denotes the query block size and $H$ is the hidden size. $L$ is the total token number and $L = S + N \times B + Q$.

After simplification:

$$\text{OP}_{\text{PEVLM}} = 2H(S^2 + Q^2 + NB^2 + QS + NQB + NSB). \tag{9}$$

As $S$, $B$, and $Q$ are fixed, both $L$ and $\text{OP}_{\text{PEVLM}}$ are linearly proportional to $N$, and therefore the computational complexity of PEVLM is $O(L)$. Compared to the $O(L^2)$ complexity of Full-Attention, PEVLM significantly reduces the computational load of the attention mechanism.

## 3.3 SINK&CONTEXT BLOCK SIZES

Since the information within each video frame represents spatial content at a specific moment, while information across frames reflects temporal dynamics, videos naturally possess structural boundaries. If the video is divided into sink and context blocks by token count, this may to some extent compromise the spatial integrity of the boundary frames shared between adjacent blocks. So we divide the video by frame.

As analyzed in the previous section, the number of frames selected for the Sink Block depends primarily on the distribution of attention scores. The optimal strategy is to include all tokens with significantly higher attention scores at the beginning of the video in the Sink Block. However, on the one hand, the number of frames containing visual tokens with a high attention score varies across different videos; on the other hand, as shown in Equation 9, a larger sink block size leads to higher latency, resulting in a trade-off between accuracy and efficiency. A similar trade-off also exists in determining the context block size. We will further analyze this in Section 5.

## 4 EXPERIMENTS

In this section, we test PEVLM from multiple perspectives against several different VLMs. The primary objective is to evaluate the accuracy, computational efficiency, and real-world applicability of PEVLM for long-video processing. And to better reflect practical deployment scenarios, we designed the experiments from two perspectives. (i) We first test with workloads of approximately 100k tokens to assess the acceleration performance of PEVLM in cloud serving platforms. (ii) Furthermore, we evaluate PEVLM on the LongVideoBench dataset under a given latency constraint, in order to measure its effectiveness in edge scenarios where both computational resources and latency are limited.

### 4.1 ACCURACY EVALUATION

This experiment aims to evaluate the impact of PEVLM on accuracy in long-video understanding tasks.

Table 1: Performance (↑) of different models and different methods on video understanding tasks evaluated at tokens from 26k to 100k.

| Method | MVBench 17s avg. | EgoSchema 3mins avg. | VideoMME < 2mins | VideoMME 2mins-1h | LongVideoBench (Val) < 1min | LongVideoBench (Val) 1min-1h | Avg. |
|---|---|---|---|---|---|---|---|
| | | | | | | | |
| _InternVL3_5-30B-A3B-Instruct_ | | | | | | | |
| Full Attn | **75.33**% | **83.60**% | **77.67**% | **63.94**% | **74.52**% | **58.81**% | **72.31**% |
| Block Attn (ICLR25) | 64.33% | 79.60% | 69.89% | 58.17% | 65.37% | 54.61% | 65.33% |
| APE (ICLR25) | 64.81% | 79.60% | 71.22% | 58.11% | 67.31% | 53.59% | 65.77% |
| Star Attn (ICML25) | 70.44% | 79.80% | 70.67% | 58.89% | 69.53% | 53.59% | 67.15% |
| PEVLM | 73.53% | 83.00% | 77.00% | 62.67% | 73.41% | 58.40% | 71.34% |
| | | | | | | | |
| _Qwen3-VL-8B-Instruct_ | | | | | | | |
| Full Attn | 69.36% | **73.00**% | **80.33**% | **68.00**% | **77.29**% | **63.52**% | **71.92**% |
| Block Attn (ICLR25) | 67.92% | 65.80% | 68.56% | 54.00% | 71.75% | 50.31% | 63.06% |
| APE (ICLR25) | 64.22% | 69.60% | 65.11% | 56.56% | 68.70% | 56.45% | 63.44% |
| Star Attn (ICML25) | 58.58% | 12.40% | 12.89% | 60.56% | 51.25% | 58.71% | 42.40% |
| PEVLM | **69.42**% | 71.40% | 80.11% | 64.17% | 76.73% | 62.09% | 70.65% |
| | | | | | | | |
| _Qwen2.5-VL-7B-Instruct_ | | | | | | | |
| Full Attn | **68.47**% | 57.00% | **74.56**% | 56.72% | 72.85% | 55.33% | 64.16% |
| Block Attn (ICLR25) | 66.03% | 47.40% | 65.22% | 50.39% | 70.64% | 47.85% | 57.92% |
| APE (ICLR25) | 66.03% | 57.20% | 50.56% | 1.06% | 54.02% | 3.38% | 38.71% |
| Star Attn (ICML25) | 60.19% | 19.80% | 3.00% | 0.00% | 49.03% | 0.00% | 22.00% |
| PEVLM | 68.44% | **61.00**% | 74.33% | **58.83**% | **74.52**% | **57.58**% | **65.78**% |
| | | | | | | | |
| _LongVILA-7B-256f_ | | | | | | | |
| Full Attn | 61.92% | 57.00% | 66.33% | 49.39% | 65.10% | 47.44% | 57.86% |
| Block Attn (ICLR25) | 57.81% | 59.00% | 64.89% | 51.44% | 61.77% | 48.05% | 57.16% |
| APE (ICLR25) | 60.00% | 60.60% | 68.56% | 51.61% | 63.71% | 46.93% | 58.57% |
| Star Attn (ICML25) | 63.39% | **61.60**% | **70.11**% | 54.17% | 63.43% | 47.44% | 60.02% |
| PEVLM | **63.53**% | 61.40% | 69.44% | **54.33**% | **65.93**% | **49.18**% | **60.64**% |
| | | | | | | | |
| _LLaVA-Video-7B-Qwen2_ | | | | | | | |
| Full Attn | **60.67**% | **58.20**% | **75.33**% | **59.99**% | **72.02**% | **55.53**% | **63.62**% |
| Block Attn (ICLR25) | 56.86% | 51.60% | 67.56% | 55.50% | 63.99% | 52.36% | 57.98% |
| APE (ICLR25) | 58.00% | 57.60% | 72.22% | 58.19% | 67.87% | 54.61% | 61.42% |
| Star Attn (ICML25) | 59.31% | 57.80% | 72.89% | 58.71% | 68.98% | 54.92% | 62.10% |
| PEVLM | 60.00% | **58.20**% | 74.33% | **59.99**% | **72.02**% | **55.53**% | 63.35% |

### 4.1.1 SETUP

We adopt LongVideoBench (Zhang & Wang, 2024), VideoMME (Fu et al., 2024), EgoSchema (Mangalam et al., 2023), MVBench (Li et al., 2024a) and MME-VideoOCR (Shi et al., 2025) as benchmarks, which are designed to comprehensively evaluate multimodal models in video understanding. All evaluations are conducted using the lmms-eval toolkit (Zhang et al., 2025b). We test on LLaVA-Video (Zhang et al., 2024c), LongVILA (Chen et al., 2024), Qwen2.5-VL (Bai et al., 2025), Qwen3-VL (Team, 2025) and InternVL3_5 (Wang et al., 2025a) as representative models. To minimize accuracy loss, all experiments are conducted with bf16 precision.

We compared the accuracy of Full-Attention, Block-Attention, APE, Star-Attention, and PEVLM in the datasets. Since the other methods are fine-tuning-free, we did not fine-tune the weights in the Block Attention test. For APE, we conducted experiments with a temperature setting of $T = 1.0$. This choice was motivated by two considerations. First, lowering the temperature further leads to additional accuracy degradation. As previously observed, parallel encoding tends to produce lower attention score distributions over context blocks in VLMs, rather than higher. Therefore, decreasing the temperature further skews the attention distribution away from that of Full-Attention, exacerbating the accuracy loss. Second, using $T = 1.0$ is favorable for large-scale deployment. Regarding the sink block (shared prefix) size, we selected the system prompt by referring to the code provided by APE. For Star-Attention, we implemented its equivalent algorithm on a single node, with the anchor size

set equal to the context block size, following the setup in the Block-Attention paper (Ma et al., 2025). All methods use a context block size of 4096 tokens. Smaller context block sizes resulted in reduced accuracy for all methods, while larger blocks significantly degraded inference performance. To ensure fair comparison with baseline methods, PEVLM adopts identical 16-frame sink and context block sizes (approximately 4k tokens for Qwen2.5-VL and LongVILA, and 3k tokens for LLaVA-Video, Qwen3-VL and InternVL3_5).

### 4.1.2 RESULTS

As shown in Table 1 and the extended results in Appendix F, PEVLM consistently achieves the highest accuracy among all efficient-attention baselines across a wide range of models, datasets and video lengths. On the latest InternVL3_5, Qwen3-VL, and Qwen2.5-VL families, PEVLM remains close to Full Attention, while Block-Attention, APE, and Star-Attention exhibit substantial drops across nearly all evaluation tracks.

The same trend holds across model scales from 2B to 32B and across both dense and MoE variants: PEVLM preserves accuracy reliably, whereas the baselines degrade significantly as model size increases or when the evaluation involves longer videos. Furthermore, the radar graph analyzes in Appendix G show that PEVLM maintains strong performance on both temporally sensitive tasks and fine-grained visual details tasks, while alternative methods exhibit inconsistent behavior across categories.

For LongVILA and LLaVA-Video, although the accuracy gaps among parallel encoding schemes are generally smaller than those in the Qwen-VL and InternVL families, PEVLM still consistently performs best. Taken together, the results in both the main table and the appendix demonstrate that PEVLM is the most stable and effective parallel encoding approach across architectures, evaluation settings, and model scales.

### 4.2 PERFORMANCE EVALUATION

**Setup** The primary objective of this experiment is to assess the improvements in computational efficiency introduced by PEVLM. We conducted computational efficiency evaluations on the Qwen2.5-VL model. We built PEVLM based on SGLang, which is a fast serving framework for large language models and vision-language models, widely used in cloud production deployment (The corresponding code will be open-sourced on GitHub once internal compliance review is fully completed). All experiments were carried out on an NVIDIA H20-96G GPGPU. Given that PEVLM primarily optimizes the attention mechanism within the LLM, we measured the execution time of both the entire LLM module and its individual attention layers separately.

**Results** As shown in Figure 4, PEVLM reduces the end-to-end inference latency of the LLM by 44% to 50%, depending on the block and sink sizes chosen. The acceleration is primarily driven by the attention module, which achieves speedups ranging from 7.79× to 33.81×, while the execution time of non-attention components remains largely unchanged.

As the sink block size or context block size increases, the speedup of the attention module gradually diminishes. This behavior is consistent with Equation 9, which indicates that the computational cost of attention grows quadratically with respect to the sizes of the sink and context blocks. Notably, because non-attention components account for a substantial portion of the total runtime, the overall end-to-end speedup degrades only moderately under larger block configurations.

However, when both context block size and sink block size increase to 16 frames, the fraction of time spent on attention rises sharply from 6.4% to 22.9%, suggesting that further enlarging the block size will lead to a more pronounced decline in overall system performance. We further analyze the trade-off between efficiency degradation and potential accuracy gains introduced by larger sink and context blocks in Section 5.

### 4.3 ACCURACY WITH LIMITED LATENCY

Deploying VLMs on edge devices presents significant challenges due to constrained computational resources and stringent latency requirements. Lacking native edge-testing infrastructure, we designed

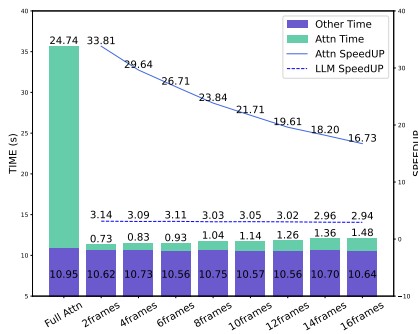

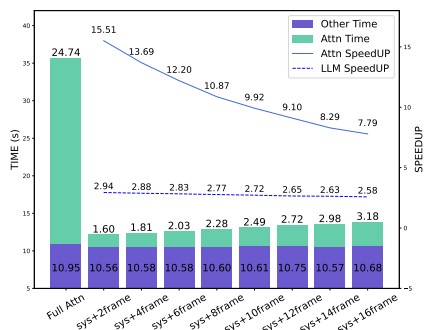

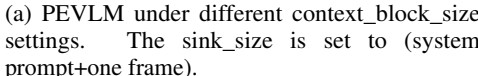

(a) PEVLM under different context_block_size settings. The sink_size is set to (system prompt+one frame).

(b) PEVLM under different sink_size settings. The context_block_size is set to (16 frames).

Figure 4: Performance of PEVLM under different block size configurations: "Attn Time" refers to attention computation time, and "Other Time" covers all other LLM costs. "Attn SpeedUP" and "LLM SpeedUP" indicate the acceleration over Attention layer and overall LLM performance.

a latency-aware simulation to evaluate the practical benefits of PEVLM by emulating real-world deployment conditions on resource-limited devices.

**Setup**  We adopt LongVideoBench as the evaluation benchmark and introduce a latency threshold during inference: Any sample that exceeds this latency limit is treated as a failure. All experiments are conducted on the NVIDIA H20-96G GPGPU. We use Qwen2.5-VL-7B-Instruct with Full-Attention as the primary baseline, and compare the performance of Full-Attention and PEVLM under identical latency constraints. To further reflect deployment in low-resource environments, where smaller models are commonly used to meet latency budgets, we also include comparisons with Qwen2.5-VL-3B-Instruct, a smaller variant Qwen2.5-VL-7B-Instruct.

Table 2: Accuracy ↑ with Limited Latency

|          | Full Attn | Full Attn (3B) | PEVLM |          |
|----------|-----------|----------------|--------|----------|
| sink     | -         | -              | sys+2f | sys+16f  |
| context  | -         | -              | 16f    | 16f      |
| No Limit | 60.43%    | 55.05%         | 61.18% | **62.23%** |
| 40s      | 59.69%    | 55.05%         | 61.18% | **62.23%** |
| 30s      | 27.08%    | 55.05%         | 61.18% | **62.23%** |
| 20s      | 23.26%    | 24.38%         | **61.03%** | 60.28% |

**Results**  As shown in Table 2, the accuracy of Full-Attention (7B) drops sharply when the latency constraint is reduced from 40s to 30s, and the smaller Full-Attention (3B) model also shows a significant decline when the limit is further tightened to 20s. In contrast, PEVLM maintains consistently high accuracy even under the strictest 20s constraint. Notably, while the larger sink size (sys+16f) achieves the best accuracy under relaxed constraints (≥30s), the smaller configuration (sys+2f) performs better under the 20s limit due to its lower latency overhead.

In general, PEVLM significantly improves accuracy in latency-constrained scenarios. Moreover, to maximize performance across diverse hardware and deployment conditions, the optimal configuration (e.g., sink/block size) should be adaptively tuned.

## 5 HOW DOES EACH COMPONENT IN PEVLM CONTRIBUTE TO THE PERFORMANCE?

Table 3 reports the ablation study across all five model families. The results show a clear and consistent trend. Adding SP yields the largest single-step improvement and forms the foundation

Table 3: Ablation study of PEVLM components

| | | | LongVideoBench | | | | | |
|---|---|---|---|---|---|---|---|---|
| SP | DF | FS | LLaVA-Video | LongVILA | Qwen2.5-VL | Qwen3-VL | InternVL3_5 | AVG |
| | | | 57.29 | 51.61 | 30.44 | 54.23 | 55.25 | 48.65 |
| ✓ | | | 57.82 | 52.58 | 59.84 | 60.21 | 57.27 | 56.88 |
| ✓ | ✓ | | 57.97 | 52.95 | 60.06 | 61.09 | 59.05 | 57.51 |
| ✓ | ✓ | ✓ | 58.94 | 53.70 | 62.15 | 64.25 | 61.52 | 59.08 |
| | | | VideoMME | | | | | |
| SP | DF | FS | LLaVA-Video | LongVILA | Qwen2.5-VL | Qwen3-VL | InternVL3_5 | AVG |
| | | | 17.56 | 57.26 | 61.11 | 54.95 | 58.63 | 49.90 |
| ✓ | | | 61.78 | 58.00 | 61.74 | 62.93 | 60.31 | 60.95 |
| ✓ | ✓ | | 62.07 | 57.30 | 62.52 | 64.99 | 64.23 | 62.22 |
| ✓ | ✓ | ✓ | 64.00 | 59.37 | 63.30 | 67.48 | 65.74 | 63.98 |

of PEVLM's effectiveness. Introducing DF offers additional but smaller gains, improving stability and accuracy across models. Incorporating FS provides the final boost, and combining all three components results in the highest accuracy on every benchmark and every model family. Overall, the three components are complementary, and the full PEVLM design provides the most robust and reliable performance in long-video settings.

## 5.1 SEQUENTIAL POSITION EMBEDDING

We further analyze the component with the greatest impact on accuracy: the preservation of sequential position embeddings. As discussed before, adopting sequential position embeddings instead of reusing position embeddings across context blocks produces attention distributions that more closely resemble those of Full-Attention, thereby aligning better with the model's expected behavior. As shown in Table 4, the use of sequential position embeddings significantly improves accuracy in both Qwen2.5-VL and LongVILA. In contrast, for the LLaVA-Video model, the benefit is marginal, likely because its attention score distribution (as shown in Figure 10 in the Appendix H) remains relatively stable even when using parallel encoding.

Table 4: Accuracy ↑ on LongVideoBench dataset with Different Position Embedding Strategy

| model | method | reuse pos | sequential |
|---|---|---|---|
| Qwen2.5-VL-7B-Instruct | APE (T=1.0) | 30.44% | 59.39% |
| | Star Attn | 29.54% | 61.41% |
| | PEVLM | 14.73% | 62.23% |
| LongVILA-7B-256f | APE (T=1.0) | 51.46% | 53.40% |
| | Star Attn | 51.76% | 53.63% |
| | PEVLM | 49.96% | 53.70% |
| LLaVA-Video-7B-Qwen2 | APE (T=1.0) | 58.19% | 58.12% |
| | Star Attn | 58.71% | 58.49% |
| | PEVLM | 58.34% | 59.01% |

## 5.2 SINK & CONTEXT BLOCK SIZE

As analyzed in the *PEVLM* section, the inference latency of PEVLM grows quadratically with the sizes of both the sink and context blocks, which aligns with the empirical results shown in Figure 4. We further investigate how these block sizes affect the accuracy of parallel encoding.

First, as noted in the *Observations* section, including the initial frames of the video in the sink is crucial. This is validated in Figure 5, where incorporating early frames into the sink leads to noticeable accuracy improvements across all models evaluated.

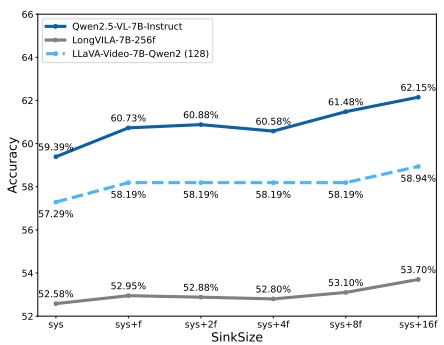 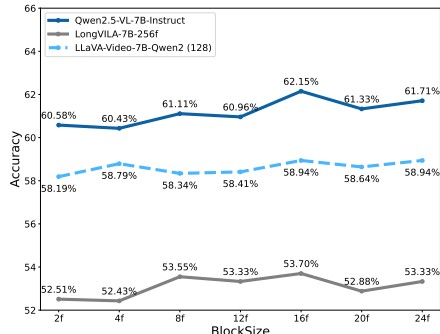

The impact of sink sizes on accuracy. The x-axis shows different sink size configurations from system prompt (sys) to system prompt plus 16 frames (sys+16f).

The impact of context block sizes on accuracy. The x-axis shows different context block size configurations from 2 frames (2f) to 24 frames (24f).

Figure 5: The impact of sink&block sizes on accuracy.

Figure 5 also shows that model accuracy generally increases with sink size. However, due to the quadratic latency growth, large sink sizes are not ideal for deployment. Therefore, a simple pre-deployment test may be necessary to determine the optimal configuration. Since there is no one-size-fits-all solution for determining the best configuration, users should make their choice based on actual requirements. For example, when latency constraints are considered, as shown in Table 2, under a 20-second latency budget, using only `[SYS] + 2 frames` as the Sink Block yields higher accuracy than `[SYS] + 16 frames`.

We also examine the effect of context block size on model accuracy. As shown in Figure 5, increasing the size of context blocks leads to consistent accuracy gains, similar to the trend observed with the sink block size. However, Figure 4 reveals that latency also increases quadratically with block size, further emphasizing the trade-off between accuracy and computational efficiency.

## 6 CONCLUSION

We introduce PEVLM, a fine-tuning-free parallel encoding strategy that significantly accelerates Vision-Language Models for long-video understanding. By preserving sequential position embeddings and leveraging a shared Sink Block, PEVLM aligns attention behavior with Full-Attention while reducing complexity from $O((T \times N)^2)$ to $O(T \times N)$. It achieves up to 7.47× attention speed-up, 44% to 50% end-to-end latency reduction, and strong accuracy gains under tight latency constraints. PEVLM offers a practical and scalable solution for multimodal real-world and long-context tasks.

## REPRODUCIBILITY STATEMENT

To ensure the reproducibility of our work, we have detailed the experimental methodology for PEVLM in Section 3, providing a clear description of its partitioning strategy, formulations, and block size considerations. Additionally, Section 4.1, Sections 4.2 and Section 4.3 further specify the comparative evaluation settings, detailing computational efficiency tests on the NVIDIA H20-96G GPGPU using the SGLang framework and latency-constrained simulations, respectively. These comprehensive details enable other researchers to replicate and validate our findings. We will also release the project code based on SGLang in the near future.

## REVISION NOTE

In response to reviewer feedback, we have significantly expanded our experimental evaluation and clarified key aspects of our method. Specifically:

- We extended experiments to include the latest InternVL3.5 and Qwen3-VL model families, covering architectures from 2B to 32B parameters (both dense and MoE), and evaluated them across five long-video benchmarks (MVBench, EgoSchema, VideoMME, LongVideoBench, MME-VideoOCR), confirming PEVLM's robustness across scales and fine-grained tasks such as video OCR;
- We conducted comprehensive ablation studies across multiple model families, demonstrating that each of PEVLM's three components—Sequential Position Embedding, Divide by Frame, and Frame Sink—consistently contributes to accuracy;
- We corrected a citation error (Line 202), fixed typographical issues throughout the manuscript, and regenerated Figures 3 and 4 in higher resolution to improve clarity.

These revisions strengthen both the empirical foundation and presentation of our work, further validating PEVLM as an effective, training-free solution for accelerating long-video inference in vision-language models.

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

APPENDIX

## A  RELATED WORK

**Parallel Encoding Mechanisms.**   Several state-of-the-art architectural improvements have been proposed to improve parallel encoding. Adaptive Parallel Encoding (APE) aligns the attention score distribution of parallel encoding with sequential attention, but its hyperparameters are highly sensitive to input samples, complicating deployment (Yang et al., 2025). Star Attention reduces prefilling latency by parallel design but is tailored for distributed systems, limiting its use in single-node and edge devices (Acharya et al., 2025). Block-Attention & TurboRAG improve accuracy through position reencoding, but require fine-tuning due to the lack of handling for attention sink phenomena (Guu et al., 2020; Lu et al., 2024). Video-XL-2 (Qin et al., 2025) summarizes historical frame information to reduce kv-cache size during video prefilling and employs chunk-based prefilling to further accelerate processing. While effective, this inevitably compresses historical content and therefore introduces potential information loss. Other methods like Native Sparse Attention and MoBA achieve similar speed-ups but require architectural changes and full retraining (Yuan et al., 2025; Lu et al., 2025).

**Efficient Attention Mechanisms.**   A growing body of work focuses on improving the efficiency of attention mechanisms for long sequences. StreamingLLM (Lab & AI, 2023) mitigates the "attention sink" phenomenon via fixed-pattern attention that selectively retains sink and sliding-window tokens. LongNet (Ding et al., 2023) introduces dilated attention, achieving linear complexity in sequence length. FlexPrefill (Lai et al., 2025) dynamically adapts attention patterns and computational budgets during inference. XAttention (Xu et al., 2025) predicts block importance via antidiagonal scoring, enabling sparsification with minimal accuracy loss. SpargeAttn (Zhang et al., 2025a) performs double-stage block-level filtering to achieve extreme prefill efficiency. These methods demonstrate the potential of sparsity- or pattern-driven attention to reduce computation, but they do not specifically target long-video prefilling scenarios, where frame-level ordering and large-scale multimodal inputs present additional challenges.

**Long-Context VLMs.**   Recent multimodal research has pushed the context length of vision-language models to hundreds of thousands of tokens. LongVA (Zhang et al., 2024a), Qwen-VL/Qwen3-VL (Team, 2025), and LongVILA (Chen et al., 2024) extend multimodal Transformers to 128K tokens with new training and architecture strategies. LongLLaVA (Wang et al., 2025b) combines Mamba and Transformer blocks for memory-efficient long-context reasoning. GIRAFFE (Pal et al., 2023) introduces optimized data pipelines and positional schemes for extreme context lengths. V2PE (Ge et al., 2024) proposes variable-position visual encodings to enhance long-range multimodal representation. Video-XL (Shu et al., 2024) and Video-XL-2 (Liu et al., 2025) develop lightweight long-video modeling frameworks based on chunked prefilling, KV-cache sparsification, and dynamic token synthesis. These advances demonstrate substantial progress in long-context multimodal modeling; however, most focus on architectural scalability or training strategies rather than efficient, accuracy-preserving parallel encoding during the prefill stage. Our work addresses this gap by designing a parallel encoding scheme tailored for long-video processing.

## B  FUTURE WORK

Although PEVLM achieves notable gains in efficiency and performance for long video tasks, several directions remain open for exploration. First, we aim to extend PEVLM to richer multimodal inputs (e.g., LiDAR, maps, IMU) commonly used in robotics and autonomous systems, which pose unique challenges for alignment and fusion. Second, PEVLM is inherently better suited for streaming and online updates, as it computes only intra-frame attention and decouples inter-frame dependencies. This design supports causal real-time processing without requiring continuous updates, substantially reducing latency in VLM/VLA systems; building on this property, adapting PEVLM to online streaming settings that process only newly arrived data within a sliding window could further enable real-time reasoning with significant speed-ups. Lastly, we plan to investigate the theoretical reasons behind the performance gains of parallel encoding over Full-Attention, potentially uncovering deeper insights for modeling long contexts.

## C    THE USE OF LARGE LANGUAGE MODELS (LLMS)

LLMs were used in this work only for spelling correction, grammar improvement, and language polishing. No content was generated by LLMs for research ideation, experimental design, data analysis, or substantive scientific contribution. All research decisions and substantive writing remain the sole responsibility of the authors.

## D    BENCHMARK DETAILS

We evaluate our method on several video understanding benchmarks that test different aspects of video comprehension:

### D.1    EGOSCHEMA [NEURIPS 2023]

EgoSchema (Mangalam et al., 2023) is a large-scale benchmark designed to evaluate multimodal Large Language Models (MLLMs) on egocentric video understanding. It consists of 100 hours of first-person video data spanning 1,270 daily activity episodes across diverse real-world environments. The benchmark introduces over 10,000 manually curated question-answer pairs, covering tasks such as object grounding, human-object interaction, activity reasoning, and intent prediction.

### D.2    MVBENCH [CVPR 2024]

MVBench (Li et al., 2024a) is a comprehensive benchmark designed to evaluate multimodal Large Language Models (MLLMs) on multi-granular video understanding. It consists of 5 task categories—including moment-level, frame-level, clip-level, video-level, and holistic video understanding—covering a wide range of temporal scopes and reasoning demands. The benchmark includes 2,562 manually annotated questions grounded in 4,119 diverse video clips, selected from real-world scenarios.

Each question is carefully designed to probe different levels of spatiotemporal understanding, from fine-grained object recognition and short-term motion tracking to long-term event inference. MVBench offers a unified and challenging evaluation protocol to assess the generalization and reasoning ability of MLLMs across granularities.

Unlike conventional third-person video datasets, EgoSchema emphasizes embodied perception and temporal reasoning from an egocentric perspective, posing unique challenges for spatial understanding, long-term memory, and causal inference in MLLMs.

### D.3    VIDEO-MME [CVPR 2025]

Video-MME (Fu et al., 2024) is a comprehensive evaluation benchmark for assessing the video understanding capabilities of multimodal Large Language Models (MLLMs). It spans 6 primary visual domains and 30 subfields, covering a diverse range of video types and temporal scenarios. The benchmark includes 900 videos with durations ranging from 11 seconds to 1 hour, totaling 254 hours of content.

To evaluate fine-grained visual and temporal reasoning, 2,700 manually annotated question-answer pairs are provided. Video-MME challenges models to comprehend both short and long video clips across different temporal granularities, making it a rigorous benchmark for evaluating the core video processing capabilities of MLLMs.

### D.4    LONGVIDEOBENCH [NEURIPS 2024]

LongVideoBench (Zhang & Wang, 2024) is a benchmark specifically designed to evaluate the long-context video understanding capabilities of multimodal Large Language Models (MLLMs). It features 1,760 videos spanning 12 diverse real-world scenarios, with video durations ranging from 5 minutes to 2 hours, totaling over 1,000 hours of content. The benchmark includes 2,400 manually annotated multi-choice questions targeting key aspects of long video comprehension.

LongVideoBench focuses on challenging long-range temporal reasoning, event tracking, and global understanding across extended video content. It aims to assess whether models can maintain coherence, memory, and attention over prolonged contexts, making it a rigorous testbed for long-form video modeling.

## D.5 MME-VIDEOOCR [NEURIPS 2025]

MME-VideoOCR (Shi et al., 2025) evaluates video-based OCR capabilities in MLLMs. It contains 1,464 videos across 44 scenarios, with 2,000 manually annotated QA pairs covering 10 task categories and 25 sub-tasks. The benchmark stresses challenges unique to video OCR, including motion blur, frame variation, and spatio-temporal text integration.

It assesses models on fine-grained text recognition, cross-frame aggregation, structured text parsing, and reasoning over dynamic visual content. Results on 18 state-of-the-art MLLMs show that video OCR remains difficult—top models reach only 73.7% accuracy. Performance notably drops on tasks requiring holistic temporal understanding or robustness against language priors, underscoring the need for high-resolution inputs and sufficient temporal coverage.

## E ABLATION STUDY OF PEVLM COMPONENTS

Table 5: Ablation study of PEVLM components (Qwen3-VL models)

| SP | DF | FS | LongVideoBench (%) | | | | | VideoMME (%) | | | | |
|---|---|---|---|---|---|---|---|---|---|---|---|---|
| | | | AVG | 2B | 4B | 8B | 32B | AVG | 2B | 4B | 8B | 32B |
| | | | 54.23 | 46.07 | 51.91 | 59.76 | 59.16 | 54.95 | 43.19 | 50.04 | 59.41 | 67.19 |
| ✓ | | | 60.21 | 55.72 | 60.58 | 61.71 | 62.83 | 62.93 | 56.26 | 61.37 | 64.04 | 70.07 |
| ✓ | ✓ | | 61.09 | 56.54 | 61.33 | 62.60 | 63.87 | 64.99 | 58.59 | 63.41 | 66.96 | 71.00 |
| ✓ | ✓ | ✓ | 64.25 | 58.49 | 64.77 | 66.49 | 67.24 | 67.48 | 60.22 | 67.11 | 69.48 | 73.11 |

Table 6: Ablation study of PEVLM components (InternVL3_5 models)

| SP | DF | FS | LongVideoBench (%) | | | | | VideoMME (%) | | | | |
|---|---|---|---|---|---|---|---|---|---|---|---|---|
| | | | AVG | 4B | 8B | 14B | 30B-A3B | AVG | 4B | 8B | 14B | 30B-A3B |
| | | | 55.25 | 53.10 | 54.90 | 55.72 | 57.29 | 58.63 | 56.41 | 57.81 | 57.81 | 62.48 |
| ✓ | | | 57.27 | 55.42 | 56.92 | 57.89 | 58.86 | 60.31 | 58.07 | 58.48 | 61.44 | 63.26 |
| ✓ | ✓ | | 59.05 | 58.49 | 58.26 | 59.01 | 60.43 | 64.23 | 62.33 | 62.70 | 65.81 | 66.04 |
| ✓ | ✓ | ✓ | 61.52 | 59.69 | 62.00 | 61.93 | 62.45 | 65.74 | 63.33 | 64.74 | 67.44 | 67.44 |

We conduct ablation experiments to quantify the contribution of the three components of PEVLM: Sequential Position alignment (SP), Divide-by-Frame (DF), and the Frame Sink mechanism (FS). Results for Qwen3-VL and InternVL3_5 are shown in Tables 5 and 6.

Across both model families, the baseline without any components yields the lowest accuracy. Introducing SP brings the largest single-step improvement. Adding DF provides further consistent gains across all model scales. Incorporating FS delivers the final boost, and the full combination (SP+DF+FS) achieves the highest accuracy in every benchmark.

Overall, the ablation results show that each component contributes positively and that all three together form a complementary and effective design for stable parallel video encoding.

## F BENCHMARK RESULTS OF QWEN3-VL AND INTERNVL3_5

Tables 7 and 8 provide a comprehensive comparison of efficient attention mechanisms across eight model checkpoints. two consistent empirical findings emerge.

Table 7: Performance (↑) of Qwen3-VL models with different methods on video understanding tasks evaluated at tokens from 26k to 100k.

| Method | MVBench 17s avg. | EgoSchema 3m avg. | VideoMME < 2m | VideoMME 2m-1h | LongVideoBench < 1m | LongVideoBench 1m-1h | VideoOCR 34s avg. | Avg. |
|---|---|---|---|---|---|---|---|---|
| Qwen3-VL-2B-Instruct | | | | | | | | |
| Full Attn | **63.28**% | 60.80% | **75.33**% | **55.94**% | 71.75% | **53.07**% | **58.26**% | **62.63**% |
| Block Attn (ICLR25) | 61.72% | 58.20% | 57.78% | 45.44% | 64.82% | 44.77% | 54.21% | 55.28% |
| APE (ICLR25) | 48.06% | 58.80% | 30.00% | 49.78% | 36.29% | 49.69% | 45.74% | 45.48% |
| Star Attn (ICML25) | 52.42% | 2.00% | 8.78% | 52.83% | 46.26% | 46.93% | 46.10% | 36.47% |
| PEVLM | 63.25% | **61.80**% | 74.11% | 53.28% | **73.13**% | **53.07**% | 58.05% | 62.38% |
| Qwen3-VL-4B-Instruct | | | | | | | | |
| Full Attn | **67.14**% | 70.60% | **79.00**% | **65.67**% | 78.67% | **62.70**% | **62.87**% | **69.52**% |
| Block Attn (ICLR25) | 65.39% | 66.20% | 63.67% | 51.22% | 74.52% | 52.25% | 60.10% | 61.91% |
| APE (ICLR25) | 53.86% | 66.80% | 43.11% | 53.50% | 45.15% | 54.41% | 45.95% | 51.83% |
| Star Attn (ICML25) | 57.22% | 11.00% | 20.22% | 57.50% | 52.08% | 57.48% | 51.49% | 43.86% |
| PEVLM | 67.11% | **71.00**% | 78.78% | 61.28% | **78.67**% | 59.63% | **62.87**% | 68.48% |
| Qwen3-VL-8B-Instruct | | | | | | | | |
| Full Attn | 69.36% | **73.00**% | **80.33**% | **68.00**% | 77.29% | **63.52**% | 64.97% | **70.92**% |
| Block Attn (ICLR25) | 67.92% | 65.80% | 68.56% | 54.00% | 71.75% | 50.31% | 61.44% | 62.83% |
| APE (ICLR25) | 64.22% | 69.60% | 65.11% | 56.56% | 68.70% | 56.45% | 58.41% | 62.72% |
| Star Attn (ICML25) | 58.58% | 12.40% | 12.89% | 60.56% | 51.25% | 58.71% | 51.90% | 43.76% |
| PEVLM | **69.42**% | 71.40% | 80.11% | 64.17% | 76.73% | 62.09% | **65.08**% | 69.86% |
| Qwen3-VL-32B-Instruct | | | | | | | | |
| Full Attn | **74.31**% | 74.80% | **83.00**% | **72.17**% | 77.01% | **64.34**% | **71.03**% | **73.81**% |
| Block Attn (ICLR25) | 70.89% | 68.80% | 72.89% | 48.17% | 72.85% | 41.50% | 67.85% | 63.28% |
| APE (ICLR25) | 70.61% | 72.60% | 75.67% | 62.94% | 74.52% | 53.48% | 65.49% | 67.90% |
| Star Attn (ICML25) | 72.36% | 3.60% | 43.67% | 59.94% | 64.54% | 44.16% | 64.87% | 50.45% |
| PEVLM | **74.31**% | **75.60**% | 82.78% | 68.28% | **77.01**% | 63.63% | 70.87% | 73.21% |

**(1) PEVLM is the only efficient method that closely matches Full Attention.** Across all models, the average accuracy gap between PEVLM and Full Attention is only **0.6–1.0 pp**, whereas BlockAttn, APE, and StarAttn fall behind by **8.0 pp**, **10.0 pp**, and **15.0 pp** on average, respectively. This trend holds for both Qwen3-VL family models and InternVL3_5 family models, demonstrating that PEVLM preserves the scaling benefits of larger backbones.

**(2) The largest gaps appear on temporal and OCR-heavy benchmarks.** In the most challenging long-range Video-MME split (2m–1h), PEVLM's deficit to Full Attention is modest (**2.2 pp** on average), while BlockAttn and StarAttn commonly lose **10–25 pp**. Similarly, on VideoOCR, PEVLM remains within < 1 **pp** of Full Attention across all checkpoints, while APE or StarAttn are typically **4–7 pp** lower.

**Conclusion.** Overall, PEVLM achieves near–Full-Attention performance across all benchmarks and model scales, while other efficient attention schemes suffer substantial degradation—particularly on temporally sensitive and OCR-centric tasks. These results underscore the effectiveness and robustness of PEVLM.

Table 8: Performance (↑) of InternVL3_5 models with different methods on video understanding tasks evaluated at tokens from 26k to 100k.

| Method | MVBench 17s avg. | EgoSchema 3m avg. | VideoMME | | LongVideoBench | | VideoOCR 34s avg. | Avg. |
|---|---|---|---|---|---|---|---|---|
| | | | < 2m | 2m-1h | < 1m | 1m-1h | | |
| InternVL3_5-4B-Instruct | | | | | | | | |
| Full Attn | **69.75**% | 63.00% | **75.00**% | **58.67**% | 71.47% | **55.84**% | 55.33% | **64.15**% |
| Block Attn (ICLR25) | 61.28% | 58.60% | 65.22% | 52.72% | 65.65% | 50.31% | 46.26% | 57.15% |
| APE (ICLR25) | 59.67% | 61.20% | 68.44% | 50.39% | 64.27% | 48.98% | 46.51% | 57.07% |
| Star Attn (ICML25) | 63.97% | 61.00% | 70.44% | 51.61% | 69.53% | 51.33% | 48.21% | 59.44% |
| PEVLM | 68.69% | **63.40**% | 74.78% | 58.00% | **72.58**% | 54.92% | **53.44**% | 63.69% |
| InternVL3_5-8B-Instruct | | | | | | | | |
| Full Attn | **70.81**% | 62.20% | **75.89**% | **60.39**% | 72.58% | **58.20**% | **56.31**% | **65.20**% |
| Block Attn (ICLR25) | 61.69% | 54.80% | 66.33% | 53.44% | 62.60% | 50.61% | 48.97% | 56.92% |
| APE (ICLR25) | 58.83% | 60.20% | 68.89% | 52.28% | 63.71% | 51.64% | 48.92% | 57.78% |
| Star Attn (ICML25) | 64.22% | 61.60% | 69.67% | 54.11% | 68.70% | 54.00% | 50.41% | 60.39% |
| PEVLM | 70.22% | **63.60**% | 75.22% | 59.50% | **72.85**% | 57.99% | 54.97% | 64.91% |
| InternVL3_5-14B-Instruct | | | | | | | | |
| Full Attn | **70.42**% | 72.60% | **77.22**% | **63.28**% | 74.52% | **58.81**% | **58.87**% | **67.96**% |
| Block Attn (ICLR25) | 62.64% | 69.00% | 68.33% | 55.67% | 65.37% | 54.61% | 51.13% | 60.96% |
| APE (ICLR25) | 49.61% | 68.00% | 68.89% | 52.28% | 67.31% | 53.59% | 50.41% | 58.58% |
| Star Attn (ICML25) | 45.36% | 70.00% | 69.67% | 54.11% | 69.53% | 53.59% | 52.97% | 59.32% |
| PEVLM | 69.69% | **72.80**% | 75.89% | 63.22% | 73.41% | 58.40% | 58.15% | 67.37% |
| InternVL3_5-30B-A3B-Instruct | | | | | | | | |
| Full Attn | **75.33**% | **83.60**% | **77.67**% | **63.94**% | 74.52% | **58.81**% | **59.33**% | **70.46**% |
| Block Attn (ICLR25) | 64.33% | 79.60% | 69.89% | 58.17% | 65.37% | 54.61% | 52.62% | 63.51% |
| APE (ICLR25) | 64.81% | 79.60% | 71.22% | 58.11% | 67.31% | 53.59% | 51.79% | 63.78% |
| Star Attn (ICML25) | 70.44% | 79.80% | 70.67% | 58.89% | 69.53% | 53.59% | 52.31% | 65.03% |
| PEVLM | 73.53% | 83.00% | 77.00% | 62.67% | 73.41% | 58.40% | 58.36% | 69.48% |

# G PER-TASK-CATEGORY ANALYSIS OF DIFFERENT PARALLEL ENCODING METHODS ON THE VIDEOMME DATASET

## G.1 BENCHMARK OVERVIEW

Video-MME is a comprehensive benchmark for evaluating long-form video understanding, specifically designed to assess both perceptual and reasoning capabilities of multimodal models over extended video sequences (Fu et al., 2024). It contains a diverse set of task categories, including *Action Recognition*, *Action Perception*, *Attribute Perception*, *Information Synopsis*, *Object Reasoning*, *Counting Problem*, *Spatial & Temporal Perception*, *Spatial & Temporal Reasoning*, and *OCR-related Problems*. These categories jointly demand strong local perception, global integration, and long-range temporal dependency modeling. As a result, Video-MME serves as a particularly suitable benchmark for studying the effectiveness of different attention and encoding mechanisms under long-context settings.

In this section, we present a detailed comparison among five representative parallel encoding mechanisms: **Full Attention** (baseline), **PEVLM**, **APE**, **BlockAttn**, and **StarAttn**. The results are illustrated in radar charts on multiple model scales, including Qwen3-VL (2B, 4B, 8B, 32B) and InternVL3.5 (4B, 8B, 14B, 30B-A3B), both dense and MoE.

## G.2 OVERALL PERFORMANCE TRENDS

Across all model families and parameter scales, Full Attention generally achieves the highest accuracy, serving as an upper-bound reference with complete token-to-token interactions. However, its quadratic complexity makes it impractical for long-context video modeling.

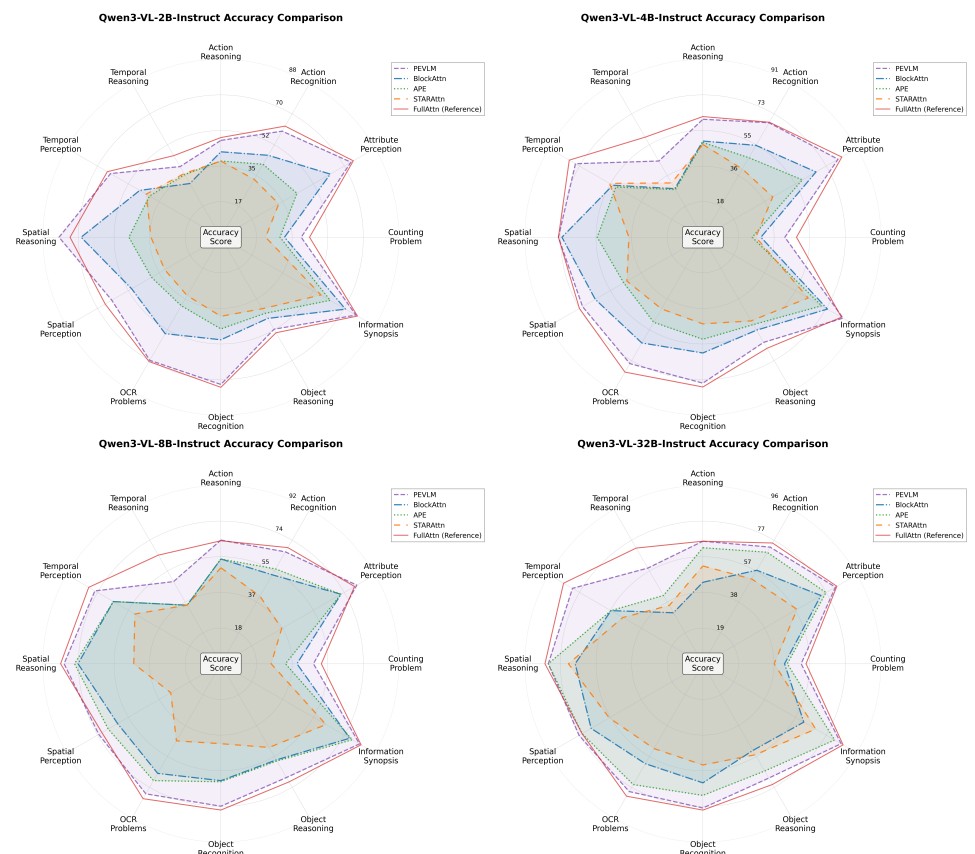

Figure 6: Radar-chart comparison of different attention mechanisms (Full Attention, PEVLM, APE, BlockAttn, StarAttn) on the Video-MME benchmark using Qwen3-VL models. PEVLM achieves consistently higher accuracy across temporally sensitive and spatially fine-grained tasks, while methods that reuse position embeddings (APE, BlockAttn, StarAttn) exhibit noticeable degradation in temporal-related and OCR-related categories.

Among the efficient alternatives, **PEVLM consistently demonstrates the closest performance to Full Attention** in almost all task categories and model scales. In contrast, both **APE** and **StarAttn** exhibit considerable performance degradation in perception-intensive and reasoning-heavy tasks, while **BlockAttn** shows moderate but unstable performance across different categories.

Notably, the relative advantage of PEVLM becomes increasingly pronounced as the model scale grows (e.g., Qwen3-VL-32B and InternVL3.5-14B/30B), suggesting that PEVLM better preserves the benefits of larger backbone models under long-sequence constraints.

### G.3 CATEGORY-WISE ANALYSIS

**Information Synopsis and Object Reasoning.** These two categories demonstrate the most significant and consistent advantages for PEVLM over other efficient methods. Both tasks require the integration of long-range semantic information across distant frames. Block-based or sparse attention structures, such as in BlockAttn and StarAttn, tend to fragment the video representation, leading to incomplete global understanding. In contrast, PEVLM's parallel encoding strategy preserves high-level temporal coherence, enabling superior global summarization and reasoning performance.

**Attribute Perception and Action Recognition.** PEVLM also closely matches Full Attention on fine-grained perceptual tasks. This indicates that its sink+divide strategy does not overly sacrifice local pixel-level or frame-level discriminative features. APE, on the other hand, shows noticeable

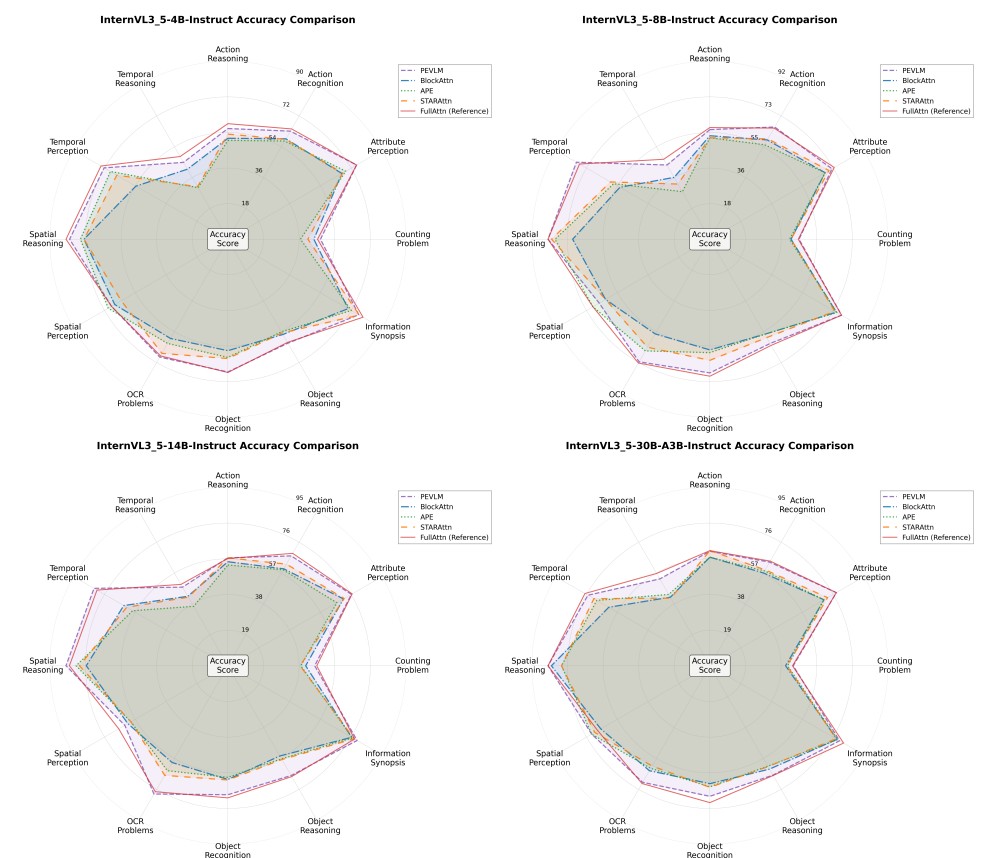

Figure 7: Radar-chart comparison of attention mechanisms on the Video-MME benchmark using InternVL3.5 models. Similar to Qwen3-VL results, PEVLM remains close to Full Attention and outperforms other parallel encoding baselines, particularly on tasks requiring strict temporal ordering and stable cross-frame alignment.

degradation in these tasks, likely due to its sensitivity to hyperparameters and unstable alignment with the original attention score distribution.

**Temporal Reasoning and Temporal Perception.** These categories highlight the difficulty of modeling long-term motion patterns and temporal dependencies. StarAttn and BlockAttn struggle to maintain cross-block temporal consistency, resulting in lower scores. In comparison, PEVLM maintains a smoother performance profile, indicating that its design better captures temporal continuity without incurring the full cost of quadratic attention.

**Spatial Reasoning and OCR Problems.** Although all efficient methods show some performance drop relative to Full Attention, PEVLM consistently ranks first among them. OCR and spatial tasks depend on both fine spatial features and long-range context (e.g., multi-frame text aggregation), which PEVLM's frame-aware parallel representation can retain more effectively.

G.4 SCALABILITY ACROSS MODEL SIZES

A critical observation from the radar plots is that PEVLM scales more favorably with increasing backbone capacity. While the relative gap between Full Attention and other methods slightly widens for APE and StarAttn at larger model sizes, PEVLM continues to closely track the Full Attention curve.

This indicates that PEVLM does not bottleneck the expressive power of larger VLMs. Instead, it allows the model to effectively leverage additional parameters even under parallel encoding constraints. Such scalability is essential for future high-capacity VLMs targeting long-video understanding.

## G.5 Key Advantages of PEVLM

From both empirical and architectural perspectives, PEVLM exhibits three major advantages:

- **High Fidelity to Full Attention:** PEVLM consistently achieves the closest approximation to Full Attention across almost every task and model size.
- **Robustness Across Categories:** Unlike APE and BlockAttn, whose performance fluctuates significantly by task type, PEVLM maintains stable superiority across both perceptual and reasoning-based categories.
- **Scalable and Training-Free:** PEVLM requires no fine-tuning or architectural modification, making it highly practical for deployment in existing VLM pipelines while still benefiting from scaling laws.

These results validate that PEVLM provides an effective and efficient alternative to Full Attention for long-context video understanding, striking an optimal balance between accuracy, scalability, and computational cost.

# H  ATTENTION SCORES DISTRIBUTION OF LLaVA-VIDEO AND LONGVILA

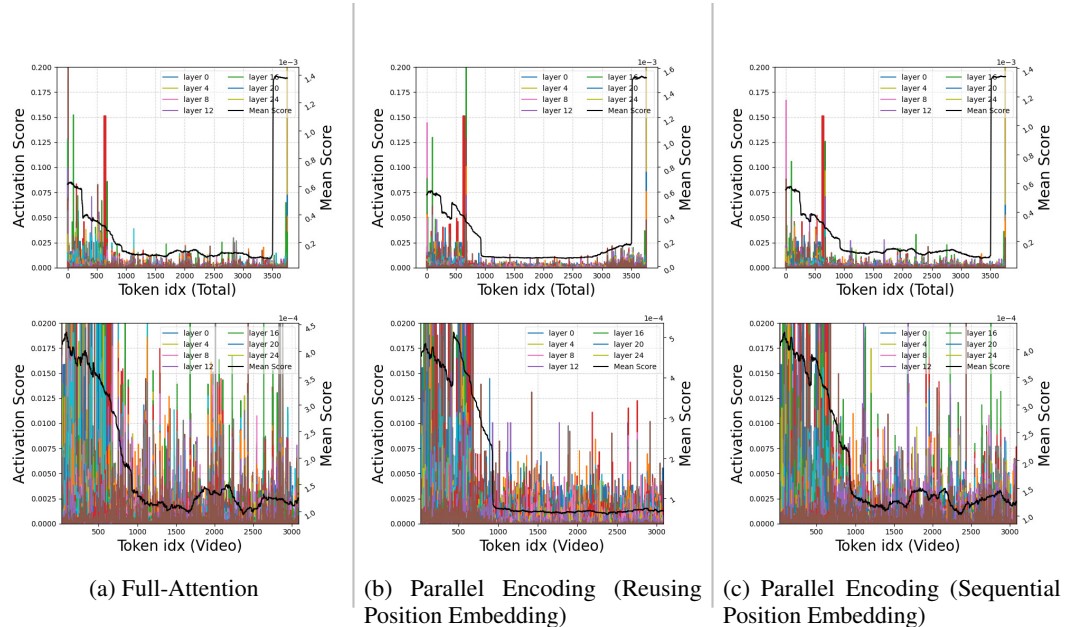

(a) Full-Attention

(b) Parallel Encoding (Reusing Position Embedding)

(c) Parallel Encoding (Sequential Position Embedding)

Figure 8: Attention Scores Distributions of Qwen2.5-VL. The top row shows the attention scores for all tokens, while the bottom row shows those for the visual tokens.

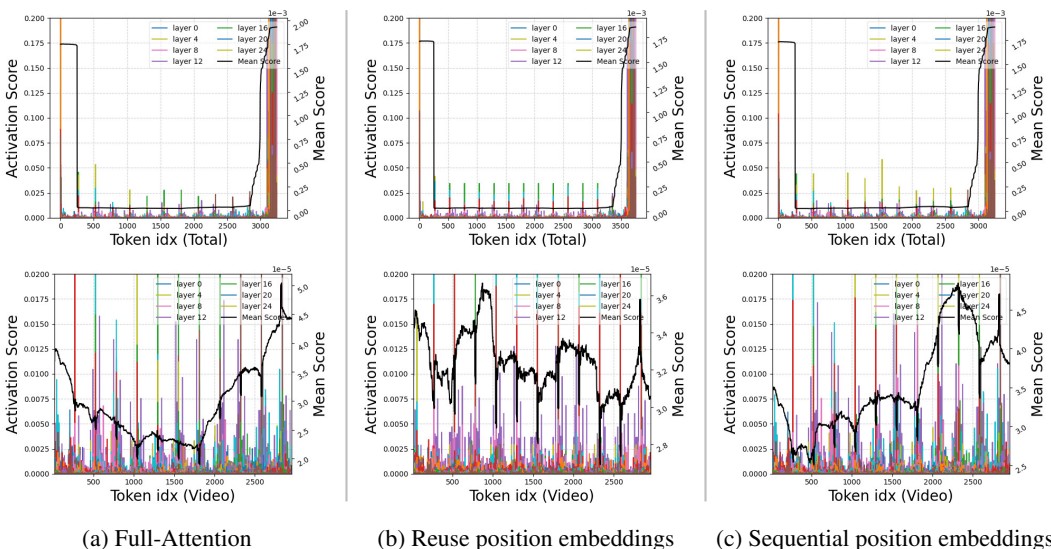

(a) Full-Attention

(b) Reuse position embeddings

(c) Sequential position embeddings

Figure 9: Attention Scores Distributions of LongVILA. The top row shows the attention scores for all tokens, while the bottom row shows those for the visual tokens.

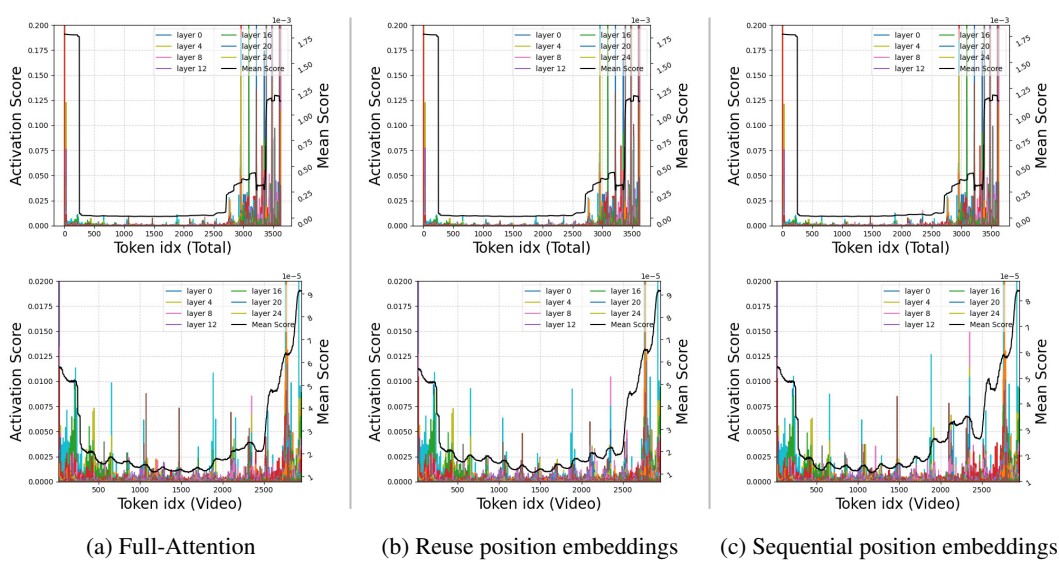

(a) Full-Attention      (b) Reuse position embeddings      (c) Sequential position embeddings

Figure 10: Attention Scores Distributions of LLaVA-Video. The top row shows the attention scores for all tokens, while the bottom row shows those for the visual tokens.

