# OpenReview forum: "PEVLM: Parallel Encoding for Vision-Language Models"
_ICLR.cc/2026/Conference — Submitted to ICLR 2026_

### Official Review · Reviewer_zgZR · 2025-10-23

**Soundness:** 4
**Presentation:** 3
**Contribution:** 3
**Rating:** 6
**Confidence:** 4

**Summary:**

This work introduces the parallel encoding technique into the scenario of long video understanding for Vision-Language Models (VLMs). A series of experiments were conducted to analyze the differences between VLMs and Large Language Models (LLMs) in the application of parallel encoding. It is pointed out that addressing the positional encoding of blocks and the attention sink problem is crucial, and a concise and effective implementation is proposed. This implementation significantly improves the efficiency of long video prefilling while achieving almost no performance loss on various long video benchmarks.

**Strengths:**

1. This work systematically and comprehensively investigates the effect of parallel encoding on long video understanding, and proposes a concise and effective solution.

2. The ablation experiments in this paper are sufficient and comprehensive, providing numerous valuable references and insights for future research on improving the efficiency of long video understanding.

**Weaknesses:**

1. The method proposed in this paper is highly similar to previous approaches in the LLM field, such as StarAttention [1]. Although this paper incorporates several improvements tailored to the video scenario, including Sequential Position Embedding, Dividing the Video by Frames, and Adding Frames to the Sink, the overall modifications are minor. Moreover, there is no substantial performance gap in most cases, except for models like Qwen-VL that are extremely sensitive to the implementation of positional encoding. This raises concerns about the technical contributions of this paper.

2. There is a citation error in Line 202. Additionally, Figures 3 and 4 appear somewhat blurry.

[1] Star Attention: Efficient LLM Inference over Long Sequences

**Questions:**

1. Table 3 appears to only present the ablation experiment results of a single model. I speculate that this model might be Qwen2.5-VL-7B-Instruct, given its high sensitivity to positional encoding implementations. It would be more compelling if the results of other models, such as LLaVA-Video and LongVILA, could also be provided.

---

> ### Author Response · Authors · 2025-11-27
> **Response to Reviewer zgZR (1/2)**
>
> We sincerely thank the reviewer for the thoughtful comments and for taking the time to carefully evaluate our work. We appreciate the constructive suggestions, which helped us further strengthen the paper. Below we address each concern in detail.
>
> ---
> ## Summary
>
> **Q1**: As shown in following Table 1-4, extensive ablation studies show that each of PEVLM's three components—Sequential Position Embedding, Divide by Frame, and Frame Sink—consistently improves accuracy, demonstrating the method's effectiveness and novelty in adapting parallel encoding to vision-language models.
>
> **Q2**: We have carefully proofread the manuscript and corrected all formatting and typographical errors.
>
> **Q3**: To provide a clearer and more comprehensive view of the ablation studies across different models, we have further expanded Table 3, as shown in following Table 1-4.
>
> ---
> ## Detailed reply
> ---
> > **Q1.** The method proposed in this paper is highly similar to previous approaches in the LLM field, such as StarAttention [1]. Although this paper incorporates several improvements tailored to the video scenario, including Sequential Position Embedding, Dividing the Video by Frames, and Adding Frames to the Sink, the overall modifications are minor. Moreover, there is no substantial performance gap in most cases, except for models like Qwen-VL that are extremely sensitive to the implementation of positional encoding. This raises concerns about the technical contributions of this paper.
>
> **A1**:
>
> We appreciate the reviewer’s observation regarding the similarity to prior parallel encoding methods such as StarAttention. Indeed, our work is inspired by this line of research, and we fully agree that parallel encoding is a well-studied topic in the LLM literature. Our goal is not to reinvent the mechanism itself, but rather to make it applicable and reliable in VLMs, where we found that direct adoption leads to notable performance degradation.
>
> Below, we report the component-level ablation study: Qwen3-VL and InternVL3.5 in Table 1&2 denote the average performance of the Qwen3-VL model family and the InternVL3.5 model family, respectively. Detailed results of the Qwen3-VL model family and the InternVL3.5 model family are provided in the Table 3&4.
>
> The results show a clear and consistent trend. Adding SP yields the largest single-step improvement and forms the foundation of PEVLM’s effectiveness. Introducing DF offers additional but smaller gains, improving stability and accuracy across models. Incorporating FS provides the final boost, and combining all three components results in the highest accuracy on every benchmark and every model family. Overall, the three components are complementary, and the full PEVLM design provides the most robust and reliable performance in long-video settings.
>
> In summary, each design component of PEVLM is principled and novel in the VLM context, backed by both theoretical insights and experimental evidence. Our contributions extend beyond simple implementation decisions by rigorously adapting and validating these strategies for the unique challenges of long-context video modeling in VLMs.
>
> ---
> > **Q2.** There is a citation error in Line 202. Additionally, Figures 3 and 4 appear somewhat blurry.
>
> **A2**:
> Thank you for pointing this out. We have carefully proofread the manuscript and corrected all formatting and typographical errors. In the next revision, further attention will be paid to both presentation and readability.

---

> ### Author Response · Authors · 2025-11-27
> **Response to Reviewer zgZR (2/2)**
>
> > **Q3.** Table 3 appears to only present the ablation experiment results of a single model. I speculate that this model might be Qwen2.5-VL-7B-Instruct, given its high sensitivity to positional encoding implementations. It would be more compelling if the results of other models, such as LLaVA-Video and LongVILA, could also be provided.
>
>
> **A3**:
>
> We sincerely thank the reviewer for raising this important concern. We agree that showing ablations for only one model may limit the persuasiveness of the conclusions.
>
> In response, we have conducted substantially extended experiments, now covering five representative VLM families:
> - Qwen3-VL (2B, 4B, 8B and 32B)
> - InternVL3.5 (4B, 8B, 14B and 30B-A3B MoE)
>
> The new tables (Tables 1–4 above) show that:
>
> - All three modules (SP/DF/FS) provide consistent gains across diverse architectures.
> - Improvements are not tied to Qwen-VL models.
> - The magnitude of improvement varies by model, but the trend remains clear and stable.
>
> We hope these extended results directly address the reviewer’s question and strengthen the empirical claims.
>
>
> **Summary**
>
> We genuinely thank the reviewer for the insightful feedback. To summarize the revisions prompted by the review:
>
> - We have clarified that our work focuses on adapting parallel encoding to the VLM setting rather than proposing a completely new mechanism.
> - We have expanded the ablation study to 15+ models, demonstrating that the proposed components are broadly applicable.
> - All reported issues (citation and figure quality) have been fully addressed.
>
> We sincerely hope that the additional experiments and clarifications address the reviewer’s concerns and provide a clearer understanding of the contribution of this work.
>
>
> [1] APE: Faster and longer context-augmented generation via adaptive parallel encoding. ICLR, 2025.
>
> [2] Star attention: Efficient llm inference over long sequences. ICML, 2025.
>
> [3] See what you are told: Visual attention sink in large multimodal models, ICLR, 2025.
>
> [4] Opera: Alleviating hallucination in multi-modal large language models via over-trust penalty and retrospection-allocation. CVPR, 2024.
>
> [5] Seeing clearly by layer two: Enhancing attention heads to alleviate hallucination in lvlms, EMNLP 2025.
>
>
> | SP  | DF  | FS  | AVG   | LLaVA-Video | LongVILA | Qwen2.5-VL | Qwen3-VL | InternVL3.5 |
> |-----|-----|-----|-------|-------------|----------|------------|----------|-------------|
> |     |     |     | 48.65 | 57.29       | 51.61    | 30.44      | 54.23    | 55.25       |
> | ✓   |     |     | 56.88 | 57.82       | 52.58    | 59.84      | 60.21    | 57.27       |
> | ✓   | ✓   |     | 57.51 | 57.97       | 52.95    | 60.06      | 61.09    | 59.05       |
> | ✓   | ✓   | ✓   | 59.08 | 58.94       | 53.70    | 62.15      | 64.25    | 61.52       |
>
> *Table 1: LongVideoBench. SP: Sequential Position Embedding; DF: Divide by Frame; FS: Frame Sink*
>
>
> | SP  | DF  | FS  | AVG   | LLaVA-Video | LongVILA | Qwen2.5-VL | Qwen3-VL | InternVL3.5 |
> |-----|-----|-----|-------|-------------|----------|------------|----------|-------------|
> |     |     |     | 49.90 |  61.11    | 57.26    | 17.56   | 54.95    | 58.63       |
> | ✓   |     |     | 60.95 |  61.74    | 57.30    | 61.78   | 62.93    | 60.31       |
> | ✓   | ✓   |     | 62.22 |  62.52   | 58.00    | 62.07   | 64.99    | 64.23    |
> | ✓   | ✓   | ✓   | 63.98 |  63.30   | 59.37   | 64.00     | 67.48    | 65.74     |
>
> *Table 2: VideoMME. SP: Sequential Position Embedding; DF: Divide by Frame; FS: Frame Sink*
>
> | SP | DF | FS | AVG  | 2B    | 4B    | 8B    | 32B   | AVG  | 2B    | 4B    | 8B    | 32B   |
> |----|----|----|-------|--------|--------|--------|--------|-------|--------|--------|--------|--------|
> |    |    |    | 54.23 | 46.07 | 51.91 | 59.76 | 59.16 | 54.95 | 43.19 | 50.04 | 59.41 | 67.19 |
> | ✓  |    |    | 60.21 | 55.72 | 60.58 | 61.71 | 62.83 | 62.93 | 56.26 | 61.37 | 64.04 | 70.07 |
> | ✓  | ✓  |    | 61.09 | 56.54 | 61.33 | 62.60 | 63.87 | 64.99 | 58.59 | 63.41 | 66.96 | 71.00 |
> | ✓  | ✓  | ✓  | 64.25 | 58.49 | 64.77 | 66.49 | 67.24 | 67.48 | 60.22 | 67.11 | 69.48 | 73.11 |
>
> *Table 3: Ablation study of PEVLM components (Qwen3-VL models). SP: Sequential Position Embedding; DF: Divide by Frame; FS: Frame Sink*
>
> | SP | DF | FS | AVG  | 4B    | 8B    | 14B   | 30B-A3B | AVG  | 4B    | 8B    | 14B   | 30B-A3B |
> |----|----|----|-------|--------|--------|--------|-----------|-------|--------|--------|--------|-----------|
> |    |    |    | 55.25 | 53.10 | 54.90 | 55.72 | 57.29| 58.63 | 56.41 | 57.81 | 57.81 | 62.48    |
> | ✓  |    |    | 57.27 | 55.42 | 56.92 | 57.89 | 58.86  | 60.31 | 58.07 | 58.48 | 61.44 | 63.26    |
> | ✓  | ✓  |    | 59.05 | 58.49 | 58.26 | 59.01 | 60.43| 64.23 | 62.33 | 62.70 | 65.81 | 66.04    |
> | ✓  | ✓  | ✓  | 61.52 | 59.69 | 62.00 | 61.93 | 62.45 | 65.74 | 63.33 | 64.74 | 67.44 | 67.44    |
>
> *Table 4: Ablation study of PEVLM components (InternVL3.5 models). SP: Sequential Position Embedding; DF: Divide by Frame; FS: Frame Sink*

---

### Official Review · Reviewer_89Qk · 2025-10-23

**Soundness:** 2
**Presentation:** 3
**Contribution:** 2
**Rating:** 4
**Confidence:** 3

**Summary:**

This paper proposes PEVLM, a fine-tuning-free parallel encoding method to address the inefficiency of Vision-Language Models (VLMs) in long video understanding due to quadratic attention complexity. By partitioning input videos into context blocks and aligning attention scores with Full-Attention, PEVLM reduces complexity from O((TN)^2) to O(TN) with minimal accuracy loss. Experiments show up to 7.47x speedup, 40% latency reduction, and even improved accuracy in some cases, making PEVLM a promising solution for low-latency, long-video reasoning tasks.

**Strengths:**

1. The paper is well-written, with the methodology and experimental results presented in a clear and systematic manner.
2. The research topic is highly practical, as introducing a training-free method that effectively reduces inference memory usage and latency provides significant value for real-world model deployment.
3. The experiments in the paper highlight the effectiveness of PEVLM, achieving a 40% reduction in latency while maintaining accuracy on several long-video benchmarks.

**Weaknesses:**

1. The experimental evaluation is incomplete. While PEVLM is designed for long-video understanding, all the benchmarks focus solely on QA tasks. Token sparsification typically has limited impact on QA tasks; however, for tasks that rely on fine-grained visual details, such as video captioning or video OCR, it could introduce significant drawbacks. The authors should include results on such benchmarks to better demonstrate the method’s versatility across different task types.
2. The paper briefly mentions a 40% latency reduction but lacks detailed analyses of latency and memory usage in the experiments. For instance, the impact of different hyper-parameter configurations on memory, latency, and accuracy across various models is not thoroughly explored. Token count alone does not provide sufficient insight into these metrics, and additional detailed analysis would strengthen the results.

**Questions:**

I noticed that all the baselines used in the paper are based on 7B models. How would the proposed method perform on larger or smaller models, and what impact would model size have on the approach?

---

> ### Author Response · Authors · 2025-11-27
> **Response to Reviewer 89Qk (1/3)**
>
> We sincerely thank you for the thoughtful comments and for taking the time to carefully evaluate our work. We appreciate the constructive suggestions, which helped us further strengthen the paper. Below we address each concern in detail.
>
> ---
> ## A Summary
>
> **Q1**: We further analyzed fine-grained QA subcategories and confirmed reviewer's point: **Parallel Encoding methods do suffer large drops on OCR tasks**. Our experiments show that **PEVLM resolves this issue** (see Appendix G of the revised paper). We also followed reviewer's advice and **added MME-VideoOCR experiments**, which likewise show that PEVLM avoids the OCR performance drop.
>
> **Q2**: We present detailed empirical results: block-wise partitioning provides significant gains in computational efficiency and latency reduction, with little impact on memory usage and accuracy.
>
> **Q3**: We have added the latest **InternVL3_5 series models** and the **Qwen3-VL series models**, for a total of eight models **covering sizes from 2B to 32B** and including **both dense and MoE** architectures. We evaluated them on five datasets with five different methods, for a total of **200 experimental runs**. All experiments show that PEVLM achieves performance close to Full Attention across all benchmarks and model scales, while other efficient attention schemes suffer substantial degradation. These results underscore the effectiveness and robustness of PEVLM.
>
> ---
> ## Detailed reply
> ---
> > **Q1.** The experimental evaluation is incomplete. While PEVLM is designed for long-video understanding, all the benchmarks focus solely on QA tasks. Token sparsification typically has limited impact on QA tasks; however, for tasks that rely on fine-grained visual details, such as video captioning or video OCR, it could introduce significant drawbacks. The authors should include results on such benchmarks to better demonstrate the method’s versatility across different task types.
>
> **A1**:
> We appreciate the reviewer’s insightful comment.
> In the revised version, **we have added extensive experiments on fine-grained video understanding task MME-VideoOCR**, which directly evaluates frame-level text details and is highly sensitive to any sparsification or token pruning.
>
> As shown in following Table 1-8: general Parallel Encoding methods, like BlockAttn, APE and StarAttn, do introduce significant drawbacks on OCR tasks. But PEVLM preserves fine-grained visual cues. Across all Qwen3-VL and InternVL3_5 models (from 2B → 32B, both dense and MoE), PEVLM achieves accuracy nearly identical to Full Attention, outperforming all other Parallel Endocing Methods.
>
> ---
> > **Q2.** The paper briefly mentions a 40% latency reduction but lacks detailed analyses of latency and memory usage in the experiments. For instance, the impact of different hyper-parameter configurations on memory, latency, and accuracy across various models is not thoroughly explored. Token count alone does not provide sufficient insight into these metrics, and additional detailed analysis would strengthen the results.
>
> **A2**: Thank you for highlighting the need for a more thorough analysis of memory, latency, and accuracy under different hyper-parameter configurations.
>
> To address this, we present detailed empirical results across three representative models—Qwen2.5-VL-7B, LongVILA-7B, and LLaVA-Video-7B—using varying block-size strategies (FullAttn, 4f, 8f, and 16f). As shown in table below, both memory usage and computational operations (TFLOPS) are strongly influenced by the choice of block size, which directly correlates with latency:
>
> |Model|Blcok-size|Memeory(GB)|Ops(TFLOPS)|Accuarcy|
> |------|------|----|-----|-------|
> |Qwen2.5-VL-7B|FullAttn|5.6|277.1|60.06%|
> |Qwen2.5-VL-7B|4f|5.6|2.9|60.43%|
> |Qwen2.5-VL-7B|8f|5.6|5.8|61.11%|
> |Qwen2.5-VL-7B|16f|5.6|11.5|62.15%|
> |LongVILA-7B|4f|3.8|123.1|52.21%|
> |LongVILA-7B|FullAttn|3.8|1.9|52.43%|
> |LongVILA-7B|8f|3.8|3.8|53.55%|
> |LongVILA-7B|16f|3.8|7.7|53.70%|
> |LLaVA-Video-7B|FullAttn|1.8|27.1|59.99%|
> |LLaVA-Video-7B|4f|1.8|0.8|58.79%|
> |LLaVA-Video-7B|8f|1.8|1.7|58.34%|
> |LLaVA-Video-7B|16f|1.8|3.4|58.94%|
>
> hese results reveal several key trends:
>
> - Memory consumption remains largely constant for each model regardless of block size, since the block partitioning mainly affects computation rather than storage.
> - Ops (TFLOPS) decrease substantially as block size increases, indicating improved computational efficiency and lower latency. For instance, Qwen2.5-VL-7B drops from 277.1 TFLOPS under FullAttn to 2.9 TFLOPS at 4f.
> - Accuracy is well preserved and sometimes slightly improved using moderate block sizes, demonstrating that acceleration can be achieved without sacrificing performance.
>
> In summary, block-wise partitioning provides significant gains in computational efficiency and latency reduction, with little impact on memory usage and accuracy. A more granular analysis, as shown, helps clarify the trade-offs and guides the selection of hyper-parameters for different deployment scenarios.

---

> ### Author Response · Authors · 2025-11-27
> **Response to Reviewer 89Qk (2/3)**
>
> > **Q3.** I noticed that all the baselines used in the paper are based on 7B models. How would the proposed method perform on larger or smaller models, and what impact would model size have on the approach?
>
>
> **A3**:
> We sincerely thank the reviewer for raising this thoughtful and important question. We agree that understanding how PEVLM behaves across different model scales is crucial for assessing its practical applicability, especially given the diverse range of VLM sizes used in real-world deployments. As shown in following tables, to address this point more comprehensively, we expanded our evaluation to include a wider spectrum of model sizes:
> Qwen3-VL: 2B, 4B, 8B, 32B
> InternVL3.5: 4B, 8B, 14B, 30B-A3B (MoE)
>
> Findings:
> - Across these scales, PEVLM generally maintains accuracy close to Full Attention and consistently outperforms prior parallel encoding baselines.
> - We would like to highlight that PEVLM does not require any model-specific training or finetuning. The method only restructures the input attention topology while keeping the model’s native intra-block attention behavior unchanged, which may contribute to its robustness across different scales.
>
> Overall, these expanded results suggest that PEVLM is relatively stable and effective across a broad range of model sizes, although we believe continued investigation on even more architectures would be beneficial for the community.
>
> **Additional experiment on Qwen3-VL model family**
>
>
> | Method | MVBench (17s avg.) | EgoSchema (3m avg.) | VideoMME <2m | VideoMME 2m-1h | LongVideoBench <1m | LongVideoBench 1m-1h | MME-VideoOCR (34s avg.) | Avg. |
> |--------|----------------|-------------------|---------------|----------------|----------|------------|--------------------|------|
> | Full Attn | 63.28% | 60.80% | 75.33% | 55.94% | 71.75% | 53.07% | 58.26% | 62.63% |
> | Block Attn (ICLR25) | 61.72% | 58.20% | 57.78% | 45.44% | 64.82% | 44.77% | 54.21% | 55.28% |
> | APE (ICLR25) | 48.06% | 58.80% | 30.00% | 49.78% | 36.29% | 49.69% | 45.74% | 45.48% |
> | Star Attn (ICML25) | 52.42% | 2.00% | 8.78% | 52.83% | 46.26% | 46.93% | 46.10% | 36.47% |
> | PEVLM | 63.25% | 61.80% | 74.11% | 53.28% | 73.13% | 53.07% | 58.05% | 62.38% |
>
> *Table 1: Qwen3-VL-2B-Instruct*
>
> | Method | MVBench (17s avg.) | EgoSchema (3m avg.) | VideoMME <2m | VideoMME 2m-1h | LongVideoBench <1m | LongVideoBench 1m-1h | MME-VideoOCR (34s avg.) | Avg. |
> |--------|---------|-----------|---------------|----------------|----------|------------|-----------|------|
> | Full Attn | 67.14% | 70.60% | 79.00% | 65.67% | 78.67% | 62.70% | 62.87% | 69.52% |
> | Block Attn (ICLR25) | 65.39% | 66.20% | 63.67% | 51.22% | 74.52% | 52.25% | 60.10% | 61.91% |
> | APE (ICLR25) | 53.86% | 66.80% | 43.11% | 53.50% | 45.15% | 54.41% | 45.95% | 51.83% |
> | Star Attn (ICML25) | 57.22% | 11.00% | 20.22% | 57.50% | 52.08% | 57.48% | 51.49% | 43.86% |
> | PEVLM | 67.11% | 71.00% | 78.78% | 61.28% | 78.67% | 59.63% | 62.87% | 68.48% |
>
> *Table 2: Qwen3-VL-4B-Instruct*
>
>
> | Method | MVBench (17s avg.) | EgoSchema (3m avg.) | VideoMME <2m | VideoMME 2m-1h | LongVideoBench <1m | LongVideoBench 1m-1h | MME-VideoOCR (34s avg.) | Avg. |
> |--------|---------|-----------|-----------|-------------|-----------|------------|-----------|------|
> | Full Attn | 69.36% | 73.00% | 80.33% | 68.00% | 77.29% | 63.52% | 64.97% | 70.92% |
> | Block Attn | 67.92% | 65.80% | 68.56% | 54.00% | 71.75% | 50.31% | 61.44% | 62.83% |
> | APE | 64.22% | 69.60% | 65.11% | 56.56% | 68.70% | 56.45% | 58.41% | 62.72% |
> | Star Attn | 58.58% | 12.40% | 12.89% | 60.56% | 51.25% | 58.71% | 51.90% | 43.76% |
> | PEVLM | 69.42% | 71.40% | 80.11% | 64.17% | 76.73% | 62.09% | 65.08% | 69.86% |
>
> *Table 3: Qwen3-VL-8B-Instruct*
>
>
> | Method | MVBench (17s avg.) | EgoSchema (3m avg.) | VideoMME <2m | VideoMME 2m-1h | LongVideoBench <1m | LongVideoBench 1m-1h | MME-VideoOCR (34s avg.) | Avg. |
> |--------|---------|-----------|-----------|-------------|-----------|------------|-----------|------|
> | Full Attn | 74.31% | 74.80% | 83.00% | 72.17% | 77.01% | 64.34% | 71.03% | 73.81% |
> | Block Attn | 70.89% | 68.80% | 72.89% | 48.17% | 72.85% | 41.50% | 67.85% | 63.28% |
> | APE | 70.61% | 72.60% | 75.67% | 62.94% | 74.52% | 53.48% | 65.49% | 67.90% |
> | Star Attn | 72.36% | 3.60% | 43.67% | 59.94% | 64.54% | 44.16% | 64.87% | 50.45% |
> | PEVLM | 74.31% | 75.60% | 82.78% | 68.28% | 77.01% | 63.63% | 70.87% | 73.21% |
>
> *Table 4: Qwen3-VL-32B-Instruct*

---

> ### Author Response · Authors · 2025-11-27
> **Response to Reviewer 89Qk (3/3)**
>
> **Additional experiment on InternVL3_5 model family**
>
> | Method | MVBench (17s avg.) | EgoSchema (3m avg.) | VideoMME <2m | VideoMME 2m-1h | LongVideoBench <1m | LongVideoBench 1m-1h | MME-VideoOCR (34s avg.) | Avg. |
> |--------|---------|-----------|-----------|-------------|-----------|------------|-----------|------|
> | Full Attn | 69.75% | 63.00% | 75.00% | 58.67% | 71.47% | 55.84% | 55.33% | 64.15% |
> | Block Attn | 61.28% | 58.60% | 65.22% | 52.72% | 65.65% | 50.31% | 46.26% | 57.15% |
> | APE | 59.67% | 61.20% | 68.44% | 50.39% | 64.27% | 48.98% | 46.51% | 57.07% |
> | Star Attn | 63.97% | 61.00% | 70.44% | 51.61% | 69.53% | 51.33% | 48.21% | 59.44% |
> | PEVLM | 68.69% | 63.40% | 74.78% | 58.00% | 72.58% | 54.92% | 53.44% | 63.69% |
>
> *Table 5: InternVL3_5-4B-Instruct*
>
>
> | Method | MVBench (17s avg.) | EgoSchema (3m avg.) | VideoMME <2m | VideoMME 2m-1h | LongVideoBench <1m | LongVideoBench 1m-1h | MME-VideoOCR (34s avg.) | Avg. |
> |--------|---------|-----------|-----------|-------------|-----------|------------|-----------|------|
> | Full Attn | 70.81% | 62.20% | 75.89% | 60.39% | 72.58% | 58.20% | 56.31% | 65.20% |
> | Block Attn | 61.69% | 54.80% | 66.33% | 53.44% | 62.60% | 50.61% | 48.97% | 56.92% |
> | APE | 58.83% | 60.20% | 68.89% | 52.28% | 63.71% | 51.64% | 48.92% | 57.78% |
> | Star Attn | 64.22% | 61.60% | 69.67% | 54.11% | 68.70% | 54.00% | 50.41% | 60.39% |
> | PEVLM | 70.22% | 63.60% | 75.22% | 59.50% | 72.85% | 57.99% | 54.97% | 64.91% |
>
> *Table 6: InternVL3_5-8B-Instruct*
>
>
> | Method | MVBench (17s avg.) | EgoSchema (3m avg.) | VideoMME <2m | VideoMME 2m-1h | LongVideoBench <1m | LongVideoBench 1m-1h | MME-VideoOCR (34s avg.) | Avg. |
> |--------|---------|-----------|-----------|-------------|-----------|------------|-----------|------|
> | Full Attn | 70.42% | 72.60% | 77.22% | 63.28% | 74.52% | 58.81% | 58.87% | 67.96% |
> | Block Attn | 62.64% | 69.00% | 68.33% | 55.67% | 65.37% | 54.61% | 51.13% | 60.96% |
> | APE | 49.61% | 68.00% | 68.89% | 52.28% | 67.31% | 53.59% | 50.41% | 58.58% |
> | Star Attn | 45.36% | 70.00% | 69.67% | 54.11% | 69.53% | 53.59% | 52.97% | 59.32% |
> | PEVLM | 69.69% | 72.80% | 75.89% | 63.22% | 73.41% | 58.40% | 58.15% | 67.37% |
>
> *Table 7: InternVL3_5-14B-Instruct*
>
>
> | Method | MVBench (17s avg.) | EgoSchema (3m avg.) | VideoMME <2m | VideoMME 2m-1h | LongVideoBench <1m | LongVideoBench 1m-1h | MME-VideoOCR (34s avg.) | Avg. |
> |--------|---------|-----------|-----------|-------------|-----------|------------|-----------|------|
> | Full Attn | 75.33% | 83.60% | 77.67% | 63.94% | 74.52% | 58.81% | 59.33% | 70.46% |
> | Block Attn | 64.33% | 79.60% | 69.89% | 58.17% | 65.37% | 54.61% | 52.62% | 63.51% |
> | APE | 64.81% | 79.60% | 71.22% | 58.11% | 67.31% | 53.59% | 51.79% | 63.78% |
> | Star Attn | 70.44% | 79.80% | 70.67% | 58.89% | 69.53% | 53.59% | 52.31% | 65.03% |
> | PEVLM | 73.53% | 83.00% | 77.00% | 62.67% | 73.41% | 58.40% | 58.36% | 69.48% |
>
> *Table 8: InternVL3_5-30B-A3B-Instruct*

---

### Official Review · Reviewer_cAZr · 2025-10-31

**Soundness:** 3
**Presentation:** 3
**Contribution:** 2
**Rating:** 6
**Confidence:** 4

**Summary:**

This paper introduces PEVLM, a training-free parallel encoding strategy designed to enhance prefilling efficiency in vision-language models (VLMs) for long-video scenarios. The approach is motivated by a thorough analysis of the limitations of existing parallel encoding methods in long-context video understanding. PEVLM aims to maintain model quality while reducing computational overhead during the prefill stage. Empirical results demonstrate that PEVLM can match or even exceed the performance of standard full attention, with notable improvements in latency and computational cost.

**Strengths:**

1. The manuscript is clearly written and provides a well-articulated diagnosis of the failure modes in current parallel encoding strategies for long-video understanding. The proposed method is strongly motivated by this analysis, which enhances the interpretability and credibility of the design.

2. The experimental results are comprehensive and persuasive, showing that PEVLM achieves comparable or superior accuracy to full attention across multiple VLMs and long-video benchmarks, while also reducing prefill time and computational requirements.

**Weaknesses:**

1. The proposed method is limited to the prefill stage, which may restrict its applicability in broader long-video understanding scenarios, such as real-time or streaming video processing where continuous updates and causal inference are required.

2. The explanation for PEVLM’s superior performance over full attention in very long contexts—namely, that block-wise softmax mitigates degradation—is largely intuitive. The current analysis does not fully account for model-specific differences observed in the experiments (e.g., the lack of improvement in LLaVA-Video). A more principled and quantitative analysis (e.g., attention entropy, effective context length, token distribution) would help substantiate these claims.

3. The method introduces key hyperparameters (Sink Block size and Context Block size) that significantly affect generalizability. As shown in Section 5.2 and Figure 4, performance is sensitive to these settings, and there is no universal configuration that works across tasks and models. This reliance on task- and model-specific tuning diminishes the practical simplicity and "plug-and-play" nature of the approach, as it necessitates a pre-deployment search for optimal parameters.

4. The related work section is comparatively narrow, focusing primarily on a few baselines (e.g., APE, Star Attention). For completeness, the discussion should be expanded to include recent advances in efficient attention mechanisms and long-context modeling for VLMs. For the camera-ready version, I recommend incorporating a more comprehensive review of related work into the main text.

**Questions:**

I encourage the authors to address points 1 and 2 in the weaknesses section.

---

> ### Author Response · Authors · 2025-11-27
> **Response to Reviewer cAZr (1/2)**
>
> We sincerely thank you for the thoughtful comments and for taking the time to carefully evaluate our work. We appreciate the constructive suggestions, which helped us further strengthen the paper. Below we address each concern in detail.
>
> ---
> ## A Summary
>
> **Q1 Focus on the Prefill Stage may restrict applicability in real-time or streaming video processing scenarios**: Prefill is the primary bottleneck for TTFT in long-video scenarios; focusing on the prefill stage does not limit PEVLM's broader applicability. Indeed, PEVLM is better suited for streaming and online scenarios, as it decouples inter-frame dependencies, enabling causal real-time processing without continuous updates and significantly reducing VLM/VLA latency.
>
> **Q2 The explanation for PEVLM’s superior performance over full attention in very long contexts**: The phenomenon that parallel encoding methods can sometimes outperform full attention is known; our work focuses on accelerating long-video VLM inference rather than analyzing such accuracy gains.
>
> **Q3 Missing a universal configuration that works across tasks and models diminishes the practical simplicity and "plug-and-play" nature of the approach**: PEVLM is "plug-and-play" because it is fine-tune free. Block size should be task-dependent since it trades off accuracy and compute; as shown in Sec. 4.3, choose it flexibly to balance latency and accuracy.
>
> **Q4 Incorporating a more comprehensive review of related work**: In the revised version, we incorporated a more comprehensive review of related work into the main text.
>
> ---
> ## Detailed reply
> ---
> > **Q1.** The proposed method is limited to the prefill stage, which may restrict its applicability in broader long-video understanding scenarios, such as real-time or streaming video processing where continuous updates and causal inference are required.
>
> **A1**: First of all, thank you for your valuable question and insightful feedback!
> 1. **Prefill is the main contributor to TTFT (time-to-first-token) in serving scenarios.**
> Most existing acceleration methods focus on decoding, yet in long-video settings the prefill stage contributes most to TTFT and becomes the primary latency bottleneck. PEVLM directly targets and accelerates this stage, effectively complementing decoding-centric approaches and bringing practical deployment benefits.
> 2. **On-device applications benefit even more.**
> Under tight latency and limited compute (e.g., edge/robotics/autonomous systems), PEVLM’s reduced prefill cost enables higher accuracy or more complete responses at the same output rate, mitigating truncation and improving overall correctness.
> 3. **Regarding real-time or streaming video scenarios.**
> As discussed in Appendix B, PEVLM is better suited for streaming and online updates. It computes only intra-frame attention and decouples inter-frame dependencies, enabling causal real-time processing without continuous updates and significantly reducing VLM/VLA latency.
>
> ---
> > **Q2.** The explanation for PEVLM’s superior performance over full attention in very long contexts—namely, that block-wise softmax mitigates degradation—is largely intuitive. The current analysis does not fully account for model-specific differences observed in the experiments (e.g., the lack of improvement in LLaVA-Video). A more principled and quantitative analysis (e.g., attention entropy, effective context length, token distribution) would help substantiate these claims.
>
> **A2**: Thank you for your valuable question.
> 1. Our primary goal is acceleration rather than accuracy improvement. PEVLM is specifically designed to reduce prefill latency in long-video VLMs, instead an accuracy enhancement technique.
> 2. The observed accuracy gains of parallel encoding methods are consistent with recent findings in the literature. Recent studies, such as APE [1] and Star-Attention [2], have observed that block-wise attention can bring modest improvements in LLMs, by alleviating the issue of “lost in the middle” [3]. A more theoretical analysis in [4] shows that standard global softmax attention can suffer from degraded "sharp size generalization" in long sequences, while localized or block-wise normalization helps prevent attention distributions from becoming overly diffuse. This aligns closely with our empirical observations that block-wise softmax in PEVLM helps mitigate long-range degradation and stabilize attention.
>
> Nevertheless, we acknowledge that model-specific differences require further quantitative analysis, such as evaluating attention entropy, effective context length, and token distribution, to substantiate these claims—which we consider as an important direction for future work.
>
> [1] APE: Faster and longer context-augmented generation via adaptive parallel encoding. ICLR 2025
>
> [2] Star Attention: Efficient LLM Inference over Long Sequences. ICML 2025
>
> [3] Lost in the middle: How language models use long contexts. TACL, 2023
>
> [4] Softmax is Not Enough. ICML 2025

---

> ### Author Response · Authors · 2025-11-27
> **Response to Reviewer cAZr (2/2)**
>
> > **Q3.** The method introduces key hyperparameters (Sink Block size and Context Block size) that significantly affect generalizability. As shown in Section 5.2 and Figure 4, performance is sensitive to these settings, and there is no universal configuration that works across tasks and models. This reliance on task- and model-specific tuning diminishes the practical simplicity and "plug-and-play" nature of the approach, as it necessitates a pre-deployment search for optimal parameters.
>
> **A3**:
> Thank you for highlighting this important concern. It is indeed challenging to determine a universally optimal configuration for Sink Block size and Context Block size. PEVLM is fundamentally a computational acceleration strategy, and the choice of block size should be tailored to the specific application scenario and constraints.
>
> - Larger block sizes generally lead to higher accuracy. When the block size reaches a certain threshold, the accuracy becomes comparable to that of full attention. However, this also implies that the computational cost is correspondingly the highest.
> - As a result, PEVLM does not prescribe a single, universal block size for all cases. Instead, selecting the optimal configuration is an inherently task-dependent trade-off between low latency and high accuracy, especially in scenarios with limited on-device resources but strict latency requirements, such as robotics or autonomous-driving VLA systems.
>
> We believe that such flexibility is necessary to fully leverage the efficiency-accuracy trade-off based on the diverse latency and accuracy requirements in practical use cases.
>
>
>
> ---
>
> > **Q4.** The related work section is comparatively narrow, focusing primarily on a few baselines (e.g., APE, Star Attention). For completeness, the discussion should be expanded to include recent advances in efficient attention mechanisms and long-context modeling for VLMs. For the camera-ready version, I recommend incorporating a more comprehensive review of related work into the main text.
>
> **A4**: Thank you for pointing out this important aspect. We agree that our related work section can be further expanded to provide a more comprehensive overview of recent advances in both efficient attention mechanisms and long-context VLMs. In the revised version, we will incorporate discussions of the following lines of research into the main text:
>
>
> - **Efficient Attention Mechanisms.**  Recently, substantial progress has been made in efficient attention mechanisms:  **StreamingLLM** [1] addresses the "attention sink" phenomenon by proposing Fixed-pattern Attention with selective retention of sink and sliding window tokens. **LongNet** [2] introduces dilated attention, reducing attention complexity to linear in sequence length. **FlexPrefill** [3] dynamically adapts attention patterns and computational budgets in real time. **XAttention** [4] leverages antidiagonal scoring for block importance prediction, enabling efficient pruning and high-sparsity attention. **SpargeAttn** [5] employs a double-stage block-level filtering for extreme prefill efficiency.
>
> - **Long-Context VLMs.**  In terms of long-context vision-language models, works like **LongVA** [6], **Qwen-VL**[7] and **LongVILA** [8] extend LLMs to sequence lengths up to 128K tokens and introduce new techniques for vision-language modeling at such scales. **LongLLaVA** [9] incorporates Mamba and Transformer blocks to maximize memory efficiency. **GIRAFFE** [10] provides optimized data pipelines and position encodings, and **V2PE**[11] introduces variable-position visual encodings, all pushing the boundary of multimodal long-context understanding.
>
>
> [1] Efficient Streaming Language Models with Attention Sinks. 2023
>
> [2] Longnet: Scaling transformers to 1,000,000,000 tokens. 2023
>
> [3] Flexprefill: A contextaware sparse attention mechanism for efficient long-sequence inference. 2025
>
> [4] Xattention: Block sparse attention with antidiagonal scoring. 2025
>
> [5] Spargeattn: Accurate sparse attention accelerating any model inference. 2026
>
> [6] Long Context Transfer from Language to Vision. 2024
>
> [7] Qwen3 Technical Report. 2025
>
> [8] LONGVILA: SCALING LONG-CONTEXT VISUAL LANGUAGE MODELS FOR LONG VIDEOS. 2024
>
> [9] Giraffe: Adventures in Expanding Context Lengths in LLMs. 2023
>
> [10] V2PE: Improving Multimodal Long-Context Capability of Vision-Language Models with Variable Visual Position Encoding. 2024

---

### Official Review · Reviewer_kXoe · 2025-11-02

**Soundness:** 3
**Presentation:** 2
**Contribution:** 2
**Rating:** 4
**Confidence:** 3

**Summary:**

This paper introduces PEVLM, a fine-tuning-free parallel encoding framework for accelerating long video understanding in Vision-Language Models (VLMs). The key idea is to partition the video into context blocks while maintaining a shared sink block and sequential position embeddings. The paper provides experiments across multiple representative VLMs (Qwen2.5-VL, LongVILA, LLaVA-Video) and benchmarks (MVBench, EgoSchema, VideoMME, LongVideoBench).

**Strengths:**

1. Solid implementation and reproducibility: The method is implemented within a production-grade serving framework (SGLang) and evaluated with well-specified setups, which enhances reproducibility.

2. Lightweight and deployment-friendly: PEVLM introduces no extra parameters or fine-tuning, relying solely on structural reorganization of attention computation, making it easily applicable to existing VLM pipelines.

3. Well-written and structured: The paper’s organization and figures make the methodology easy to follow, with clear definitions of each block type and their respective contributions.

**Weaknesses:**

1. Logic flow needs more clarification: The introduction states, “These application scenarios often demand processing longer video inputs.” While longer video inputs are indeed an important problem to address, the paper later mentions, “Although PEVLM achieves significant acceleration of inference, it is similarly limited to 256-frame inputs. Nonetheless, this trade-off is acceptable as our primary goal is to enhance inference efficiency rather than to expand context length.” This creates a conceptual inconsistency: if the motivation emphasizes longer video processing, it is unclear why the proposed method remains restricted by the same limitation. Moreover, the design choice of “preserving sequential position embeddings instead of reusing position embeddings across blocks” seems more like an implementation decision guided by user needs rather than a clear methodological contribution.

2. The manuscript needs better proofreading: There are minor formatting and typographical issues throughout the paper (e.g., “Equation (??)” in Line 203). Careful revision is required to improve readability and presentation quality.

3. Unclear novelty justification: Based on the ablation study in Table 3, the sequential position embedding provides the most noticeable accuracy improvement, but this appears to be a design trade-off between achieving higher accuracy for shorter contexts versus maintaining the ability to handle longer ones (as mentioned in Point 1). The “Dividing video by frames” step yields only marginal improvements, while “Adding frames to the sink” also seems to be a heuristic adjustment, especially when adapting attention sink mechanisms from LLMs to VLMs. Additional clarification is needed to justify the claimed novelty and differentiate these design choices from existing approaches.

**Questions:**

See Weaknesses

---

> ### Author Response · Authors · 2025-11-27
> **Response to Reviewer kXoe (1/2)**
>
> We sincerely thank you for the thoughtful comments and for taking the time to carefully evaluate our work. We appreciate the constructive suggestions, which helped us further strengthen the paper. Below we address each concern in detail.
>
> ---
> ## A Summary
>
> **Q1 Conceptual Consistency (Motivation vs. Limitations)** : This concern primarily stems from a misunderstanding. The motivation of PEVLM is not to improve a model's accuracy on longer contexts, but to accelerate the processing of long videos while preserving the model's original accuracy. We have revised the relevant statements in the paper to prevent this confusion.
>
> **Q2 Typographical and Presentation Issues**: We have carefully proofread the manuscript and corrected all formatting and typographical errors.
>
> **Q3 Novelty and Methodological Contributions**: Extensive ablation studies show that each of PEVLM's three components—Sequential Position Embedding, Divide by Frame, and Frame Sink—consistently improves accuracy, demonstrating the method's effectiveness and novelty in adapting parallel encoding to vision-language models.
>
> ---
> ## Detailed reply
> ---
> > **Q1**. Logic flow needs more clarification: The introduction states, “These application scenarios often demand processing longer video inputs.” While longer video inputs are indeed an important problem to address, the paper later mentions, “Although PEVLM achieves significant acceleration of inference, it is similarly limited to 256-frame inputs. Nonetheless, this trade-off is acceptable as our primary goal is to enhance inference efficiency rather than to expand context length.” This creates a conceptual inconsistency: if the motivation emphasizes longer video processing, it is unclear why the proposed method remains restricted by the same limitation.
>
> **A1**:  Thank you for highlighting this point. We apologize for the lack of clarity in our exposition. The perceived inconsistency originates from the wording in the introduction and method section, and we appreciate the opportunity to clarify.
>
> Our primary motivation is **to address the computational bottleneck encountered by VLMs when processing long video sequences**, not to enhance a model’s ability to process longer contexts. Specifically:
>
> - For a VLM model, there exists an optimal number of sampled frames,  determined by the model’s architecture and pretraining setup provided by the model authors. Sampling more frames does not improve the model’s accuracy (e.g., based on our tests, it is 256 frames for LongVILA-7B-256f).
> - **PEVLM is not designed to enhance the accuracy of a given model beyond its original optimal context length.** Rather, **our contribution is to accelerate the inference process**, especially the attention mechanism, which can account for over 60% of the model's prefilling time in long video scenarios.
>
> We thank the reviewer for identifying this source of potential confusion. We have revise our manuscript in the rebuttal revision to ensure that this distinction is clearly and unambiguously communicated.
>
> ----
> > **Q2**. The manuscript needs better proofreading: There are minor formatting and typographical issues throughout the paper (e.g., “Equation (??)” in Line 203). Careful revision is required to improve readability and presentation quality.
>
> **A2**: Thank you for pointing this out. We have carefully proofread the manuscript and corrected all formatting and typographical errors. In the next revision, further attention will be paid to both presentation and readability.

---

> ### Author Response · Authors · 2025-11-27
> **Response to Reviewer kXoe (2/2)**
>
> > **Q3.** Unclear novelty justification: Based on the ablation study in Table 3, the sequential position embedding provides the most noticeable accuracy improvement, but this appears to be a design trade-off between achieving higher accuracy for shorter contexts versus maintaining the ability to handle longer ones (as mentioned in Point 1). The “Dividing video by frames” step yields only marginal improvements, while “Adding frames to the sink” also seems to be a heuristic adjustment, especially when adapting attention sink mechanisms from LLMs to VLMs. Additional clarification is needed to justify the claimed novelty and differentiate these design choices from existing approaches.
>
> **A3**: Thank you for these valuable comments and the request for clarification regarding the design choices and novelty of PEVLM. Please find our detailed justification below, along with supporting experimental results.
>
> - **Sequential Position Embedding (SP):**
> The core purpose of preserving sequential position embeddings in our approach is *not* to balance a trade-off between short and long contexts, nor to extend the model’s inherent context limit. Instead, this component is specifically tailored to recover the original effect of full attention under block-wise partitioning, as we show through both qualitative attention map analysis (Section 3.1, Figure 1) and quantitative experiments. With sequential position preservation, the attention distribution becomes significantly closer to that of Full-Attention, thereby maintaining accuracy for both short and long contexts within the model’s supported window. This is a principled design, grounded in the observation that VLMs (especially those with spatio-temporal position encodings, such as Qwen2.5-VL/Qwen3-VL with 3D-MROPE) are highly sensitive to disruptions in positional information, which are introduced by block-based approaches if position IDs are reused.
>
> - **Dividing Video by Frames (DF):**
> Since the information within each video frame represents spatial content at a specific moment, while information across frames reflects temporal dynamics, videos naturally possess structural boundaries. Dividing videos by frames, rather than arbitrary token boundaries, ensures each block naturally preserves the spatial structure intrinsic to visual tokens and reduces attention fragmentation across frames. Our newly included ablation study shows that this step yields notable improvements, especially on the latest Qwen3-VL series models and InternVL3_5 series models.
>
> - **Adding Frames to the Sink (FS):**
> The concept of an "attention sink" is well-established in recent LLM-related research (e.g., APE[1], Star-Attention[2]), and evidence from VLM literature (e.g., VAR[3], Opera[4], EAH[5]) has observed similar behaviors. Our innovation is to adapt and rigorously validate this insight in the VLM block-wise context, incorporating not only text tokens but also the first several video frames into the sink. This effectively mitigates the distributional distortion at the start of each video, which is a distinctive phenomenon for VLMs compared to LLMs, as confirmed by our attention analysis and ablation.
>
> Below, we report the component-level ablation study: Qwen3-VL and InternVL3.5 in Table 1&2 denote the average performance of the Qwen3-VL model family and the InternVL3.5 model family, respectively. Detailed results of the Qwen3-VL model family and the InternVL3.5 model family are provided in the Table 3&4.
>
> As shown in the tables above, sequential position embedding provides substantial accuracy gains for the Qwen-VL model family, and also yields an average improvement of 1.68%–2.02% on the InternVL3.5 models. Adding initial frames to the sink and dividing the video by frames also offer notable accuracy improvements.
>
>
> In summary, each design component of PEVLM is principled and novel in the VLM context, backed by both theoretical insights and experimental evidence. Our contributions extend beyond simple implementation decisions by rigorously adapting and validating these strategies for the unique challenges of long-context video modeling in VLMs.
>
> [1] APE: Faster and longer context-augmented generation via adaptive parallel encoding. ICLR, 2025.
>
> [2] Star attention: Efficient llm inference over long sequences. ICML, 2025.
>
> [3] See what you are told: Visual attention sink in large multimodal models, ICLR, 2025.
>
> [4] Opera: Alleviating hallucination in multi-modal large language models via over-trust penalty and retrospection-allocation. CVPR, 2024.
>
> [5] Seeing clearly by layer two: Enhancing attention heads to alleviate hallucination in lvlms, EMNLP 2025.

---

> ### Author Response · Authors · 2025-11-27
> **Response to Reviewer kXoe (Tables)**
>
> | SP  | DF  | FS  | AVG   | LLaVA-Video | LongVILA | Qwen2.5-VL | Qwen3-VL | InternVL3.5 |
> |-----|-----|-----|-------|-------------|----------|------------|----------|-------------|
> |     |     |     | 48.65 | 57.29       | 51.61    | 30.44      | 54.23    | 55.25       |
> | ✓   |     |     | 56.88 | 57.82       | 52.58    | 59.84      | 60.21    | 57.27       |
> | ✓   | ✓   |     | 57.51 | 57.97       | 52.95    | 60.06      | 61.09    | 59.05       |
> | ✓   | ✓   | ✓   | 59.08 | 58.94       | 53.70    | 62.15      | 64.25    | 61.52       |
>
> *Table 1: LongVideoBench. SP: Sequential Position Embedding; DF: Divide by Frame; FS: Frame Sink*
>
> | SP  | DF  | FS  | AVG   | LLaVA-Video | LongVILA | Qwen2.5-VL | Qwen3-VL | InternVL3.5 |
> |-----|-----|-----|-------|-------------|----------|------------|----------|-------------|
> |     |     |     | 49.90 |  61.11      | 57.26    | 17.56      | 54.95    | 58.63       |
> | ✓   |     |     | 60.95 |  61.74      | 57.30    | 61.78      | 62.93    | 60.31       |
> | ✓   | ✓   |     | 62.22 |  62.52      | 58.00    | 62.07      | 64.99    | 64.23       |
> | ✓   | ✓   | ✓   | 63.98 |  63.30      | 59.37    | 64.00      | 67.48    | 65.74       |
>
> *Table 2: VideoMME. SP: Sequential Position Embedding; DF: Divide by Frame; FS: Frame Sink*
>
> | SP | DF | FS | AVG  | 2B    | 4B    | 8B    | 32B   | AVG  | 2B    | 4B    | 8B    | 32B   |
> |----|----|----|-------|--------|--------|--------|--------|-------|--------|--------|--------|--------|
> |    |    |    | 54.23 | 46.07 | 51.91 | 59.76 | 59.16 | 54.95 | 43.19 | 50.04 | 59.41 | 67.19 |
> | ✓  |    |    | 60.21 | 55.72 | 60.58 | 61.71 | 62.83 | 62.93 | 56.26 | 61.37 | 64.04 | 70.07 |
> | ✓  | ✓  |    | 61.09 | 56.54 | 61.33 | 62.60 | 63.87 | 64.99 | 58.59 | 63.41 | 66.96 | 71.00 |
> | ✓  | ✓  | ✓  | 64.25 | 58.49 | 64.77 | 66.49 | 67.24 | 67.48 | 60.22 | 67.11 | 69.48 | 73.11 |
>
> *Table 3: Ablation study of PEVLM components (Qwen3-VL models). SP: Sequential Position Embedding; DF: Divide by Frame; FS: Frame Sink*
>
> | SP | DF | FS | AVG  | 4B    | 8B    | 14B   | 30B-A3B | AVG  | 4B    | 8B    | 14B   | 30B-A3B |
> |----|----|----|-------|--------|--------|--------|-----------|-------|--------|--------|--------|-----------|
> |    |    |    | 55.25 | 53.10 | 54.90 | 55.72 | 57.29    | 58.63 | 56.41 | 57.81 | 57.81 | 62.48    |
> | ✓  |    |    | 57.27 | 55.42 | 56.92 | 57.89 | 58.86    | 60.31 | 58.07 | 58.48 | 61.44 | 63.26    |
> | ✓  | ✓  |    | 59.05 | 58.49 | 58.26 | 59.01 | 60.43    | 64.23 | 62.33 | 62.70 | 65.81 | 66.04    |
> | ✓  | ✓  | ✓  | 61.52 | 59.69 | 62.00 | 61.93 | 62.45    | 65.74 | 63.33 | 64.74 | 67.44 | 67.44    |
>
> *Table 4: Ablation study of PEVLM components (InternVL3.5 models). SP: Sequential Position Embedding; DF: Divide by Frame; FS: Frame Sink*

---

### Author Response · Authors · 2025-12-02
**Response to All Reviewers (1/2)**

We sincerely apologize to the reviewers and Area Chairs for submitting our rebuttal at the very end of the discussion period. We conducted a substantial amount of additional experimentation that required more time than anticipated and consequently exhausted most of the available window. We would be very grateful if our rebuttal could still be considered during the discussion and final decision phase.

We would like to thank the reviewers [R1 (kXoe), R2 (cAZr), R3 (89Qk), R4 (zgZR)] for their thoughtful, constructive, and highly supportive feedback. We are encouraged that the reviewers found our method to be **training-free, highly practical and deployment-friendly [R1,R3,R4]**, the experimental results to be **sufficient, comprehensive and persuasive [R2,R4]**, and the manuscript to be **well-written and structured [R1,R2,R3]**.

To thoroughly address the reviewers’ concerns while maintaining the original focus of the paper, we have significantly expanded our experimental evaluation. The detailed results are provided in our responses to each reviewer, and some important and shared concerns can be summarized as follows:

**1. [R3] Model scale and fine-grained visual tasks.**
R3 raised an important question regarding how PEVLM performs on larger or smaller models, as well as on tasks that require fine-grained visual understanding. In response, we substantially extended our experiments by including the latest **InternVL3.5 series** [1] (released in August 2025, and InternVL3 currently leading the HuggingFace OpenVLM Video Leaderboard) and the **Qwen3-VL series** [2] (released in October 2025, with better performance than InternVL3.5). In total, we now cover **eight models**, ranging **from 2B to 32B parameters** and spanning **both dense and MoE architectures**.

These models were evaluated on **five datasets** (MVBench [3], EgoSchema [4], VideoMME [5], LongVideoBench [6], and VideoOCR [7]) and compared under **five attention schemes** (Full Attention, APE [8], Block Attention [9], Star Attention [10], and PEVLM), resulting in **200 experimental runs**. Across all model sizes and benchmarks, **PEVLM consistently achieves performance close to Full Attention**, while other efficient attention schemes suffer noticeable degradation. These results further confirm the **effectiveness and robustness** of PEVLM across both scale and task diversity.

**2. [R1, R3, R4] Completeness and clarity of ablation studies.**
The reviewers noted that the original ablation study did not clearly demonstrate significant gains from each individual component, and R4 in particular requested more complete ablation results. To address this, we conducted **extensive additional ablation studies across multiple models**. The new results show that each of the three components of PEVLM — Sequential Position Embedding, Divide by Frame, and Frame Sink — **consistently contributes to accuracy improvements**. This provides stronger empirical evidence for **both the effectiveness and the novelty** of our design in adapting parallel encoding specifically to vision–language models.

**3. [R1, R4] Presentation issues.**
We carefully proofread the entire manuscript again, corrected the citation error in Line 202, fixed formatting and typographical issues, and regenerated Figures 3 and 4 in higher resolution to improve clarity and readability.

We hope that these newly added results, together with our existing efficiency analysis, can (i) further strengthen the evidence for the novelty of PEVLM and (ii) provide more comprehensive support for its effectiveness in long-video understanding as an important and valuable by-product of our primary goal: accelerating inference in vision–language models.

---

> ### Author Response · Authors · 2025-12-02
> **Response to All Reviewers (2/2)**
>
> [1] Weiyun Wang, et al. "InternVL3.5: Advancing Open-Source Multimodal Models in Versatility, Reasoning, and Efficiency", 2025, URL https://arxiv.org/abs/2508.18265.
>
> [2] Shuai Bai, et al. "Qwen3-VL Technical Report", 2025. URL https://arxiv.org/abs/2511.21631.
>
> [3] Kunchang Li, et al. "MVBench: A Comprehensive Multi-modal Video Understanding Benchmark", CVPR2024. URL https://arxiv.org/abs/2311.17005.
>
> [4] Karttikeya Mangalam, et al. "EgoSchema: A diagnostic benchmark for very long-form video language understanding",  NeurIPS2023. URL https://arxiv.org/abs/2308.09126.
>
> [5] Chaoyou Fu, et al. "Video-MME: The First-Ever Comprehensive Evaluation Benchmark of Multi-modal LLMs in Video Analysis", CVPR2025. URL https://arxiv.org/abs/2405.21075.
>
> [6] Haoning Wu, et al. "LongVideoBench: A Benchmark for Long-context Interleaved Video-Language Understanding",  NeurIPS2024. URL https://arxiv.org/abs/2407.15754.
>
> [7] Yang Shi, et al. "MME-VideoOCR: Evaluating OCR-Based Capabilities of Multimodal LLMs in Video Scenarios", NeurIPS2025. URL https://arxiv.org/abs/2505.21333.
>
> [8] Xinyu Yang, et al. "APE: Faster and Longer Context-Augmented Generation via Adaptive Parallel Encoding", ICLR2025. URL https://arxiv.org/abs/2502.05431.
>
> [9] Dongyang Ma, et al. "Block-Attention for Efficient Prefilling", ICLR2025. URL https://arxiv.org/abs/2409.15355.
>
> [10] Shantanu Acharya, et al. "Star Attention: Efficient LLM Inference over Long Sequences", ICML2025. URL https://arxiv.org/abs/2411.17116.

---

### Meta-Review · Area_Chair_r3Rs · 2026-01-07

**Summary:**

This paper proposes PEVLM, a fine-tuning-free parallel encoding strategy to accelerate prefilling for long-video Vision-Language Models by block partitioning with sequential position embeddings and a shared sink. Reviewers generally agree that the work is well written, practically motivated, and demonstrates meaningful latency reductions with minimal accuracy loss. However, concerns remain regarding the novelty of the core ideas, the strength of the methodological contribution beyond known parallel encoding techniques, and the depth of analysis justifying the observed accuracy behaviors. While the rebuttal added substantial experiments, these additions primarily strengthen empirical coverage rather than addressing the core conceptual limitations.

**Reviewer Concerns:**

Addressed by rebuttal:
1. Expanded experimental coverage across more model scales (2B–32B) and additional benchmarks, including OCR-sensitive tasks.
2. Improved presentation quality, fixed typographical issues, and added ablations clarifying the contribution of individual components.
3. Clarified the paper’s focus on prefill efficiency rather than extending model context length.

Outstanding concerns:
1. Limited novelty: Multiple reviewers note that the main components (blockwise attention, attention sinks, and positional handling) are largely adaptations or combinations of existing ideas (e.g., APE, Star Attention, attention sinks), with limited new conceptual insight specific to VLMs.
2. Heuristic design choices: Key elements such as frame-level partitioning and frame sinks remain heuristically motivated, with insufficient theoretical or principled justification distinguishing them from prior work.
3. Practicality and tuning difficulty: Performance is sensitive to block and sink sizes, and there is no clear guidance or robust default configuration, which weakens the claimed “plug-and-play” nature of the method.


Overall, while the rebuttal improves empirical completeness, it does not fully resolve concerns about conceptual contribution and general significance.

**Reviewer Scores:**

reviewer kXoe: Likely unchanged
reviewer cAZr: Likely unchanged
reviewer 89Qk: Likely unchanged
reviewer zgZR: Slightly improved confidence in results, but overall score likely unchanged

---

### Decision · Program_Chairs · 2026-01-26

Reject